



# Proxies and uncertainties for $^{13}$C/$^{12}$C ratios of atmospheric reactive gases emissions

Sergey Gromov[1,2], Carl A. M. Brenninkmeijer[1] and Patrick Jöckel[3]

[1] Max Planck Institute for Chemistry, Mainz
[2] Institute of Global Climate and Ecology (Roshydromet and RAS), Moscow
[3] Deutsches Zentrum für Luft- und Raumfahrt (DLR), Institut für Physik der Atmosphäre, Oberpfaffenhofen, Weßling

*Correspondence to:* Sergey Gromov (sergey.gromov@mpic.de)

**Abstract.** We provide a comprehensive review of the proxy data on the $^{13}$C/$^{12}$C ratios and uncertainties of emissions of reactive carbonaceous compounds into the atmosphere, with a focus on CO sources. Based on an evaluated setup of the EMAC model, we derive the isotope-resolved dataset of its emission inventory for the 1997−2005 period. Additionally, we revisit the calculus required for the correct derivation of uncertainties associated with isotope ratios of emission fluxes. The resulting overall surface CO emission δ$^{13}$C in 2000 of −(25.2±0.7)‰ is in line with the previous bottom-up estimates and a factor of two less uncertain. In contrast to this, we find that uncertainties of the respective inverse modelling estimates are substantially larger due to the correlated nature of their derivation. We reckon the δ$^{13}$C values of surface emissions of higher hydrocarbons being within −24‰ to −27‰ (uncertainty typically below ±1‰), with an exception of isoprene emissions being close to −30‰. The isotope signature of ethane surface emission coincides with earlier estimates, however integrating very different source inputs. δ$^{13}$C values are reported relative to V-PDB.

## 1 Introduction

[1] Next to the kinetic chemistry implementation, emissions of airborne compounds constitute perhaps the most crucial aspect of a modelling system dealing with the chemical state of Earth's atmosphere. A consistent emission setup, in turn, requires (i) a careful selection of the emission inventories, (ii) adequate approaches to special cases (*e.g.*, boundary conditions for the long-lived species) and, not less important, (iii) estimates of the pertinent uncertainties. The latter, typically being largest in comparison to the other sources of error in the model (such as for instance reaction rate coefficients), are customarily disregarded, when the resulting simulated mixing ratios are reported. Often the inferred variation (temporal or spatial) of the species' abundance is quoted, which, however, does not represent an adequate uncertainty estimate. The situation complicates, if the isotope-resolved emissions (*i.e.*, fluxes separated using the information on the isotope ratios of the emitted compounds) are to be used. For instance, which factors determine a particular emission source isotope ratios? How do the latter (and their respective uncertainties) influence the uncertainties of the underlying fluxes? At last, what is the contribution of the emissions uncertainties to the simulated mixing/isotope ratios' overall uncertainties, and how comprehensive the model implementation should be to provide this information?



[2] The above mentioned issues and questions interested us in the course the implementation of a fully $^{12}C/^{13}C$-resolved comprehensive trace gas atmospheric chemistry study with the **ECHAM/MESSy Atmospheric Chemistry** (EMAC) model (Jöckel *et al.*, 2006; Jöckel *et al.*, 2010), particularly for the stable carbon isotope extension of its emission setup, which we communicate in this paper. The reader is referred to the preceding phases of this model development, *viz.* the isotope exten-
sion of the kinetic chemistry submodel MECCA (**M**odule **E**fficiently **C**alculating the **C**hemistry of the **A**tmosphere) and its application to simulating the carbon and oxygen isotope composition of gas-phase constituents within the CAABA (**C**hemistry **A**s **A B**oxmodel **A**pplication) atmospheric box-model (Sander *et al.*, 2011; Gromov *et al.*, 2010). Both EMAC (which embodies an atmospheric chemistry general circulation model, AC-GCM) and CAABA serve as base models within the **M**odular **E**arth **S**ubmodel **Sy**stem (MESSy, Jöckel *et al.*, 2005) we employ. The overarching aim of our studies is a con-
sistent simulation of the isotopic composition of atmospheric carbon monoxide (CO) in a detailed and more comprehensive (in comparison to previous attempts, see Sect. 4) framework of the EMAC model, which we will communicate in subsequent papers. In addition to CO, the current study provides a bottom-up assessment of the emission $^{13}C/^{12}C$ isotope ratios for the suite of other carbonaceous compounds, the information that we believe will be useful for other isotope-enabled (modelling) studies focussing on these.

[3] The manuscript consists of three main parts. In the first part (Sect. 2), we briefly reiterate the implementation of the trace gas emissions in the evaluation setup of the EMAC model (MESSy Development Cycle 2, Jöckel *et al.*, 2010, referred hereafter to as "EVAL$_2$") and supplement it with the formulation of the emission fluxes' isotope separation we introduce. Furthermore, we derive the practical approaches for calculating combined flux/isotope ratio uncertainties of emissions in Sect. 2.2. The second part (Sect. 3) revisits proxies for signatures ($^{13}C/^{12}C$ isotope ratios) of particular emission sources for
CO, non-methane hydrocarbons (NMHCs), biogenic volatile organic (VOCs) and other carbonaceous compounds represented by EMAC. Special focus is on CO (the tracer of our primary interest) and its precursors. Finally, in the last part (Sect. 4) we summarise the results and discuss our estimates in comparison with previous studies. We recapitulate our results in Sect. 5 with concluding remarks.

## 2 Emission processes in EMAC

[4] The emission of trace gases in EMAC is treated by the submodels OFFEMIS (formerly OFFLEM), ONEMIS (formerly ONLEM) and TNUDGE, which embody **off-l**ine and **on-l**ine **em**ission processes, and a pseudo-emission approach (**t**racer **nudg**ing), respectively, as detailed by Kerkweg *et al.* (2006). The conventional way of accounting for emissions applied in EMAC is realised by adjusting the tendencies of a given tracer, or, optionally in case of surface emissions, by modifying its vertical diffusive flux boundary conditions at the lowest model layer.

[5] The off-line emission process embodies a prescribed (pre-calculated) tracer flux into the atmospheric reservoir at the surface layer(s) or, for instance for the emission from air transportation sector, at respective altitudes. This type of emission



does not require a parameterisation dependent on the model parameters. The EVAL$_2$ setup includes the emissions from datasets comprising the following categories:

– anthropogenic emissions, based on the EDGAR emission inventory (detailed in Sect. 3.1),

– biomass burning emissions (GFED project database, 2$^{nd}$ version, see Sect. 3.2), and

– biogenic emissions based on the OLSEN/GEIA databases (see Sect. 3.3, respectively).

Various key assumptions determine the emission isotopic signatures. Depending on the specificity of the emission category, each of the datasets requires separate pre-processing for the isotopic extension. These are described in Sects. 3.1 to 3.5, respectively.

[6] The on-line emissions, in contrast, are calculated during the runtime and require some of the model variables (*e.g.* surface temperature or precipitation) for calculating the resulting emission flux at the given model time step. For example, online emission suits for parameterisation of the trace gas emissions related to the biosphere-atmosphere interaction processes. In particular, the EVAL$_2$ setup includes the online emissions of VOCs (isoprene/monoterpenes) from plants (see below, Sect. 3.3.1), which were scaled to achieve the net yearly emissions of $305-340$ Tg(C) of isoprene, respectively (see Poz-
zer *et al.*, 2007, Supplementary Material). Upon this adjustment, more realistic mixing ratios of isoprene in the boundary layer are achieved in EMAC simulations.

[7] At last, the pseudo-emission approach (TNUDGE) is a technique performing the relaxation (nudging) of the mixing ratios of sufficiently long-lived tracers towards prescribed (in space/time) fields. These can be, for example, the zonally averaged tracer gradients compiled from ground observations. In the EVAL$_2$ setup, the mixing ratios of $CH_4$, chlorinated carbons
($CH_3CCl_3$, $CCl_4$, $CH_3Cl$) and $CO_2$ are prescribed as the lower boundary conditions using the observed mixing ratios. The isotopic separation of these pseudo-emission fields for carbonaceous species is described below in Sect. 3.5.

[8] Further details of the emission processes implementation in EMAC and the corresponding model parameterisations are given by Kerkweg *et al.* (2006), Jöckel *et al.* (2006), Pozzer *et al.* (2007), Pozzer *et al.* (2009) and Jöckel *et al.* (2010). In the next sections we describe chiefly the choice of the isotope emission signatures for the model setups including the stable car-
85 bon isotope configuration.

## 2.1 Isotopic separation of the fluxes

[9] The isotopic extension procedure consists of the separation of the regular (*i.e.*, sum of the abundant and rare isotope bearing) species fluxes into the individual isotopologues fluxes accounting for the given isotopic ratio and thus the isotope content of a given species. Additionally, the consistency between the regular flux and the sum of isotopically separated fluxes is
90 verified. The rare isotopologues fluxes are calculated by weighting the regular species flux with the respective fractions $^{rare,if}$ according to





$$ ^{\mathrm{rare},i}f = \frac{^{\mathrm{rare},i}R \cdot q}{1 + 1 - q \cdot \sum_j {}^{\mathrm{rare},j}R}, \quad {}^{i}R = \delta^{i} + 1 \cdot {}^{i}R_{\mathrm{st}} . \tag{1} $$

Here, $q$ is the number of atoms of the selected isotope in a given species' molecule, $^{i}R$ is the isotopic ratio of a particular isotope $i$ in the flux, $^{i}R_{\mathrm{st}}$ is the reference standard isotope ratio, respectively. When accounting for multiple rare isotopes, all ratios are required for the correct calculation of the resulting fraction of each of the isotopologues. The abundant isotopologue flux fraction, in turn, is calculated as

$$ ^{\mathrm{abun}}f = 1 - \sum_j {}^{\mathrm{rare},j}f , \tag{2} $$

thus assuring that the sum of isotopically separated fluxes of the abundant and rare isotopologues equals the regular flux value. The resulting fluxes of the regular species and its isotopologues are:

$$ \begin{cases} ^{\mathrm{abun}}F = F \cdot {}^{\mathrm{abun}}f \\ ^{\mathrm{rare},i}F = F \cdot {}^{\mathrm{rare},i}f \\ F \equiv {}^{\mathrm{abun}}F + \sum_j {}^{\mathrm{rare},j}F \end{cases} \tag{3} $$

For the sake of clarity, the molecular fractions $f$ above are calculated plainly from the atomic content $q$ and the isotopic ratios. The isotopic compositions of the emission fluxes, nevertheless, are conventionally (and within this study) reported using delta values $\delta^{i}$, which relate the isotope ratio $^{i}R$ and the standard ratio $^{i}R_{\mathrm{st}}$ in (1). For the emission $\delta^{13}C$ values (or emission "signatures") the V-PDB scale with $^{13}C R_{\mathrm{st}}$ of $11237.2 \times 10^{-6}$ (Craig, 1957) is used. We note that this value is nominally outdated since the last re-determination of the carbon isotope ratio of the NBS 19 reference material used to define the "hypothetical" V-PDB scale introduced after the former PDB primary material was exhausted (see Chapter 40 in de Groot, 2004, also Zhang *et al.*, 1990). Owing to the differences between the former (*i.e.*, assigned from PDB) and revised scales, a change in isotope composition corresponding to 1‰ in $\delta^{13}C$ on the PDB-scale is about 0.001176 per mil larger on the V-PDB scale, which implies *ex post facto* different absolute abundances derived using the same $\delta^{13}C$ values reported. The resulting emission $\delta^{13}C$ signatures presented here are sensitive to the choice of these standards, since all emission fluxes are defined through them. Nonetheless, errors introduced by adopting outdated values are negligible compared to uncertainties introduced by the other factors, *e.g.* laboratory/model estimates of the emission strengths and signatures, as we show below in Sect. 3.6.

[10] During the isotopic extension of the emission data, the preparation tools import the regular emission fields (usually provided in netCDF format (http://www.unidata.ucar.edu/software/netcdf) with the fluxes values in units of molecules $m^{-2} s^{-1}$), process these according to the given isotopic signatures and output fields containing the individual isotopologue fluxes. These in turn are read in by the model data import interface and utilised in a conventional way by the emission submodels (*e.g.*, OFFEMIS). Depending on the source data used, the spatial resolution of the emission datasets varies. The input fields





are transformed to the model grid during the model integration with the help of the NCREGRID submodel (Jöckel, 2006),
which provides the consistent (flux-conserving) re-gridding algorithm.

## 2.2  Emission uncertainties analysis

[11] It is desirable to estimate the uncertainties associated with the emission signatures for the subsequent analysis of the
modelling results, particularly in view of comparison with observational data. However, deriving the isotope composition
uncertainties for composites of the various different sources with superposed individual isotopic ratios is an intricate task.
First, it should be clearly comprehensible how the uncertainties of the isotopic ratios are related, particularly in view of
summing of several compartments (*e.g.* emission fluxes from different sources) of various isotope mixtures, all given with
their individual uncertainties for the abundance and isotope composition. Second, the uncertainties associated with the
*amounts* being summed are expected to influence the combined uncertainty of the *ratio* of the final aggregate, as a conse-
quence of the law of error propagation. To give an example, even if the isotopic signature of each share (*i.e.* particular emis-
sion type) is determined (ideally) absolutely precisely, the non-zero uncertainties associated with the amounts of each share
(*i.e.*, emission fluxes) impose a non-zero uncertainty on the final isotopic signature of the total (emission). The approaches to
calculate combined emission and its isotope composition uncertainties are only sparingly documented in the literature, there-
fore they are derived below. The following analysis is based on the common practical fundament of uncertainties as de-
scribed, for instance, by Drosg (2009) and by Criss (1999).

[12] Foremost, it is expedient to switch from using the relative isotopic composition to the actual equivalent ratio, *i.e.* from $\delta^i$
to $^iR$. The use of delta variables would introduce impermeable complexities in subsequent calculations, because in contrast to
ratios, it is much more difficult to relate delta-values to extensive quantities such as fluxes. The relation of the uncertainty
$\langle \delta^i \rangle$ reported for the delta value $\delta^i$ to the uncertainty $\langle ^iR \rangle$ of the corresponding ratio $^iR$ is

$$\langle {}^iR \rangle = \left( \frac{d\delta^i}{d\,{}^iR} \right) \Delta \delta^i = {}^iR_{st} \cdot \langle \delta^i \rangle \,. \tag{4}$$

Here and further, the notation from Eqs. (1)−(3) is applied. For clarity the angle brackets $\langle\ \rangle$ are introduced in place of con-
ventional "Δ" to denote the uncertainty values. The delta-value uncertainty is linearly proportional to the ratio uncertainty
with the reference standard ratio being the proportionality factor. Further, the ratio $^iR$ can approximate the relation of the $i$th
rare isotopologue influx $^{\mathrm{rare},i}F$ to the total (regular) emission flux $F$ as

$$^iR = \frac{^{\mathrm{rare},i}F}{^{\mathrm{abun}}F + \dfrac{q-1}{q}\displaystyle\sum_j {}^{\mathrm{rare},j}F} \simeq \frac{^{\mathrm{rare},i}F}{F} \,, \tag{5}$$

assuming that the fraction of the rare isotopologues is negligibly small in the total flux, which is valid for the isotopes of the
light elements (*e.g.* C, N, O). This is the only approximation that affects the further analysis. Neglecting the abundant iso-



topes in the rare isotopologues introduces errors in the estimate of $F$ on the order of $q \cdot 1\,\%$ for carbonaceous species, assuming an average fraction of $^{13}$C carbon of 1 % in the total flux. Thus the resulting approximation of the flux

$$^{\text{rare},i}F \simeq {}^{i}R \cdot F \tag{6}$$

is approximately 1 % inaccurate for CO and 5 % for isoprene ($C_5H_8$), *i.e.* depending on the number of carbon atoms incorporated in the species molecule. Compared to the typically large errors for the emission fluxes (see below), this inaccuracy is an order of magnitude smaller.

[13] The total emission flux $F_e$ is an integral of the particular emission source fluxes $F_s$. Employing the same notation, the total regular (sum of rare and abundant) and rare isotopologue emission fluxes are

$$
\begin{aligned}
F_e &= \sum_s F_s, \\
^{\text{rare},i}F_e &= \sum_s {}^{\text{rare},i}F_s = \sum_s {}^{i}R_s \cdot F_s = {}^{i}R_e \cdot F_e
\end{aligned}
\tag{7}
$$

The summation in Eq. (7) is performed over the emission sources using index $s$. Clearly then, the resulting total flux isotopic ratio $^{i}R_e$ is

$$^{i}R_e = \varphi \sum_s {}^{i}R_s \cdot F_s, \qquad \varphi \equiv \left( \sum_s F_s \right)^{-1} \tag{8}$$

Here, $\varphi$ is introduced for the sake of notation simplification. Noteworthy, in Eq. (8) the source fluxes $F_s$ have to be used, but not the total flux $F_e$, since every $F_s$ contributes to the total uncertainty with the individual uncertainty $\langle F_s \rangle$.

[14] It is important for the applied method to differentiate whether or not the uncertainties associated with individual emission fluxes' magnitudes and/or isotope ratios are *correlated*, that is, the various given estimates depend on each other. Examples of such are inverse modelling and other "top-down" approaches which intrinsically correlate the fluxes from different emission sources by fitting their (isotope mass-balanced) sum to the given integral. The "bottom-up" estimates, on the contrary, are typically derived using independent proxies (*e.g.*, country fuel usage statistics, satellite-derived mass of burned matter). Of course, uncertainties of guesses (*e.g.*, if the emission comes predominantly from a particular plant material characterised by the distinct isotope signature) cannot be accounted for. The combined uncertainty accounting for the error propagation is calculated with the total differential of the function describing the product, in forms which are different for the correlated and uncorrelated cases. Thus, the combined uncertainty $\langle F_e \rangle$ of the total emission $F_e$ in Eq. (7) expressed through the uncertainties of correlated (inferred "top-down") components $\langle F_s \rangle$ of individual sources $F_s$ is

$$\langle F_e \rangle = \sum_s \left| \frac{\partial F_e}{\partial F_s} \right| \cdot \langle F_s \rangle = \sum_s \langle F_s \rangle \tag{9}$$





*i.e.*, a simple (linear) addition of the individual uncertainties. In the case of uncorrelated (estimated "bottom-up") total flux components, the resulting combined uncertainty is derived using the quadratic form of Eq. (9), which yields the square root of the sum of squared components $\langle F_s \rangle$, respectively:

$$\langle F_e \rangle = \sqrt{\sum_s \langle F_s \rangle^2} \ . \tag{10}$$

Analogously, the combined uncertainty $\langle R_e \rangle$ for the resulting total emission ratio $R_e$ is calculated from both, flux components ($F_s \pm \langle F_s \rangle$) and ratio components ($R_s \pm \langle R_s \rangle$), as (index $n$ varies similarly to $s$, enumerating the sources)

$$\langle {}^i R_e \rangle = \sum_s \left( \left| \frac{\partial {}^i R_e}{\partial F_s} \right| \cdot \langle F_s \rangle + \left| \frac{\partial {}^i R_e}{\partial R_s} \right| \cdot \langle {}^i R_s \rangle \right) =$$
$$= \sum_s \left( \left| \varphi^2 \cdot \sum_n F_n \ {}^i R_s - {}^i R_n \right| \cdot \langle F_s \rangle + \left| \varphi \cdot F_s \right| \cdot \langle {}^i R_s \rangle \right) \tag{11}$$

for the correlated case. The first term of the final sum in Eq. (11) describes the uncertainty in the isotope ratio arising purely from the uncertainty in emission strengths modified by the difference in the isotopic ratios between each pair of sources. The second term adds the sources' isotope ratio uncertainties weighted by the corresponding emission fluxes. In the case of uncorrelated estimates, the quadratic form of Eq. (11) yields the square root of a similar expression incorporating the above-mentioned terms squared:.

$$\langle {}^i R_e \rangle = \sqrt{\sum_s \left( \left( \varphi^2 \cdot \sum_n F_n \ {}^i R_s - {}^i R_n \right)^2 \cdot \langle F_s \rangle^2 + \varphi \cdot F_s{}^2 \cdot \langle {}^i R_s \rangle^2 \right)} \ . \tag{12}$$

Eqs. (9)−(12) in their form can be employed for the uncertainty estimation of any given combination of isotopic compartments, referring only to their abundances and isotopic ratios. Thus, it is not necessary to account for the ratios of the other rare isotopes (*cf.* isotopologue fraction calculation in Eq. (1)). We remark here that using Eqs. (9)−(12) implies that the final combined uncertainties have the normal distribution about their mean values (*i.e.*, standard deviations), despite that such may not be the case for individual emission flux estimates. Under the assumption of symmetricity for all individual uncertainties involved, however, normally distributed $\langle R_e \rangle$ will be indeed the consequence of the law of uncertainty propagation (see D'Agostini, 2004 for details).

# 3 Proxies and ${}^{12}$C/${}^{13}$C ratios of emissions

## 3.1 Anthropogenic emissions

[15] The anthropogenic emissions in EVAL$_2$ are based on the EDGAR database (version 3.2 "Fast Track 2000" (32FT2000), van Aardenne *et al.*, 2005) as detailed by Pozzer *et al.* (2007). This inventory was compiled for the year 2000. It is notewor-





thy that, despite its complex structure (the emission is distributed to tens of various categories, or "sectors"), the database has no seasonality, *i.e.* spatially distributed emission fluxes composing the emission are constant throughout the year. The inventory comprises approximately 40 sectors referring to the different anthropogenic emission sources (summarised in Table 1), which enables to assign characteristic isotopic signatures individually to each sector. The influx is distributed to the surface and multiple near-surface model layers, depending on the emitted species and the emission sector. This serves to account for specific sources that deliver the pollutants to the various effective altitudes. The majority of sectors (except for power generation, industrial fuel usage and waste treatment) are associated with the surface and adjacent layers representing 45 m and 140 m heights. The sources from remaining sectors are represented with the various plume updrafts distributed to the higher layers (spanning from 240 m to 800 m above the ground). The detailed anthropogenic emission setup and vertical distribution of the emission heights is described by Pozzer *et al.* (2009).

[16] Table 1 lists the carbon isotopic signatures for CO and emitted compounds assigned to the particular sector for anthropogenic emissions. Unfortunately, to date the information in the literature on the measured isotopic compositions of the different emitted compounds is scarce, particularly for NMHCs and other VOCs. Therefore, here the choice for the unknown signatures will follow the EDGAR categorisation, assuming the emission source material (*e.g.* crops, bio- or fossil fuels) and its characteristic processing (generally either biomass burning or high-temperature combustion) to determine the resulting isotopic ratio of the emitted tracer.

[17] The least uncertain signature is for fossil fuel usage, most of which is on account of the transportation sectors. It is associated with an average characteristic composition of $-27.5\,‰$ in $\delta^{13}C$, as reported for the world average engine exhaust by Stevens *et al.* (1972) and used as a proxy value for CO and other NMHCs/VOCs. Although quite diverse emitted CO isotope signatures were measured for various engine/fuel types (Kato *et al.*, 1999a), any better assessment based on these signatures is not feasible, because the inventory does not provide the related information. The average value from Stevens *et al.* (1972), nonetheless, agrees with more recent estimates. Thus, from measurements of CO isotopic composition in two cites in Switzerland, Saurer *et al.* (2009) infer the $\delta^{13}C$ signature of the transportation source of $-(27.2\pm1.5)\,‰$, contrasting heavier CO emitted from local wood combustion sources. A similar transportation-emitted CO $\delta^{13}C$ average value ensues from the observations in a Swiss highway tunnel study by Popa *et al.* (2014), *viz.* $-(27.5\pm0.6)\,‰$ (the average $\pm2\sigma$ of the two Keeling plot-derived source $\delta^{13}C$ signatures from the tunnel entrance and exit data is quoted).

[18] Statistically insignificant variability in emission isotope ratios for transportation-related sources of selected NMHCs has been reported by Rudolph *et al.* (2002) with the signatures for the majority of species equating to within the measurement precision of $2\,‰$ that of CO mentioned above. The exception of significant enrichment was found for ethyne ($C_2H_2$), which is not represented in the MECCA chemistry mechanism (as of $EVAL_2$ setup) and may potentially constitute an enriched, however, very moderate source (see, for example, Ho *et al.*, 2009). This is somewhat coherent with $^{13}C$ enrichments found to accompany ethyne formation during the burning process (Czapiewski *et al.*, 2002). Altogether it is generally recognised that the fossil-related sources reflect the average isotopic ratios of the precursor crude oils. The aircraft emissions are associated



with this source as well. However, the corresponding EDGAR emission (class F57) is replaced by the inventory compiled by

Schmitt and Brunner (1997) in EVAL$_2$.

[19] In analogy to the fuel combustion category (sectors "F"), the same isotopic signature ($-27.5$‰) is used for the industrial category (sectors "I"). It is expedient to assume that those sources represent dominantly the fossil nature of the precursor carbon, as the emission is mainly associated with the combustion of fuels in the majority of the industrial processes. An example is iron and steel production (sector I10), where CO is emitted concomitantly during the thermal processing of the product

in the furnaces (IISI, 2004). On the other hand, the influence of industrial sectors on the resulting emission signature should be minor, taking into account their small share in the overall anthropogenic emission. The comparison of the contributions of each EDGAR sector in case of CO emission is presented in Fig. 1. Notably, the largest fluxes are associated with sectors B40 (biofuel consumption in the residential/commercial sector) and F51 (non-$CO_2$ combustion emissions from road transport), thus the input shares of these two sectors are decisive for the overall isotopic composition of CO in EDGAR. The total emis-

sion associated with industrial sectors amounts to 34.5 Tg(CO) yr$^{-1}$, that comprises approximately 6.3 % of the total anthropogenic source.

[20] The less certain isotope signatures are associated, in turn, with the biofuel use (sectors "B") because of large uncertainties associated with the source influx estimates and somewhat unclear definition of this category itself. Although we reckon that "biofuel use" in EDGAR refers to predominantly combustion of fuel wood and vegetable oils, the category includes industri-

al activities that may imply usage of fuels (*e.g.,* liquid, gas, solid) *produced* from biomass (Olivier *et al.*, 2002). To eliminate a potentially wrong association with the biofuel category, we discuss the isotope signatures of the woodfuel and waste/residue crops sources under the "biomass burning" category below. We remark that this activity comprises likely the major fraction of the "biofuel use" emissions related to heating and cooking in Asian and African regions (Yevich and Logan, 2003). No detailed information is available about the biofuel production and use in other regions, however, particularly

for the period the EDGAR inventory was compiled for. Likewise, there are no specific measurements of the isotopic signatures of CO and other NMHCs/VOCs from biofuel sources reported yet (Goldstein and Shaw, 2003). These mainly comprise the use (primarily by combustion) of vegetable oil- and biomass-derived fuels, of which biodiesel and ethanol constitute the major parts (Demirbas, 2008). Although ethanol is included in the "biofuel combustion" category in EDGAR, neither the proportion of ethanol/biodiesel fuel sources nor the origin of precursor biogenic material is reflected in the inventory. A

rough estimate of the isotopic signature is feasible nonetheless, assuming a certain average composition of the source biomass and negligible isotope effects accompanying the emission. On average, plant material is enriched in $^{13}C$ with respect to fossil fuels and can be considered as a composite of the carbon originating from two cardinal kinds of plant species, namely $C_3$ and $C_4$ plants (explained in detail in the following, see Sect. 3.3.1). Briefly, the isotopic compositions of those differ conspicuously owing to the differences in the photosynthesis mechanisms, yielding typical compositions of $-27$‰ for $C_3$ plants

and $-12$‰ for $C_4$ plants (see, *e.g.*, Dawson *et al.*, 2002). The expected composition of the mixture is hence constrained by these values. Within the current study we follow Emmons *et al.* (2004) and adopt the value of $-25$‰, which corresponds to



an approximate 4:1 ratio of $C_3$ to $C_4$ plant material. There are, however, estimates that report a significant fraction of $C_4$ plants being used in global biofuel production. Thus, O'Connor (2009) quote the source plants species used for ethanol and biodiesel production. Whilst biodiesel is mainly produced from $C_3$ species like soy, rapeseed, canola and oil palm tree, etha-
nol is predominantly manufactured from corn and sugarcane, which are $C_4$ crops. Projecting this partitioning on the gross production rates for the year 2000 (Demirbas, 2009) of $156 \cdot 10^8$ and $9.7 \cdot 10^8$ litres for ethanol and biodiesel, respectively, will yield a rather high value for the average emission signature of $-12.9\,‰$ for these fuels. Here, the fractionation associated with the fermentation process during the ethanol production is assumed to be negligible, although a few studies (Vallet *et al.*, 1998; Zhang *et al.*, 2003) indicate that the biogenic ethanol may be even slightly enriched with respect to the source material. A substitution of the reference biofuel $\delta^{13}C$ signature of $-25\,‰$ with the above derived value of $-12.9\,‰$ will result in an unlikely strong increase (greater than $+8\,‰$) in the overall surface CO emission $\delta^{13}C$ in East Asia and Central Africa, compared to that for Europe and North America ($+1.6\,‰$ and $+1.1\,‰$, respectively), where bio-petrol is being more extensively used. The sensitivities to such substitution for the $\delta^{13}C$ of NMHCs/VOCs emissions are lower, *viz.* $+4.9\,‰$ (East Asia) and $+2.8\,‰$ (Central Africa) *vs.* $+1.0\,‰$ and $+0.8\,‰$ for Europe and North America, respectively. This rough analysis suggests that the sensitivity of simulated CO and NMHCs $\delta^{13}C$ to biofuel $^{13}C/^{12}C$ signature for Europe and North America will be likely below the (rather large) uncertainties associated with the biofuel category emission fluxes and isotope ratios itself (see also Sect. 3.6 below).

[21] The original biomass burning emission inventory of the EDGAR database (referring to land use, sectors "L") in the current setup is substituted by the more comprehensive GFED inventory described in the following section, with the exception of the agricultural waste burning sector (L43), which is not included in GFED. The emission $\delta^{13}C$ signature of $-22.2\,‰$ is assigned to this source using the average composition of the burned material estimated for 2000 by Randerson *et al.* (2005). They use the $C_3/C_4$ ratio of the burned vegetation inferred with the help of a vegetation-inclusive inversion-adjusted model and comparison with observed $CO_2$ isotope ratios. A different signature of $-21.3\,‰$ for CO is used, following the estimation similarly based on plant distribution, fuel loads and neglecting concomitant fractionations as described by Conny (1998). The estimates of burned plant composition by Randerson *et al.* (2005) do not consider the potential kinetic isotope effects that may escort biomass burning emission for various tracers.

[22] Czapiewski *et al.* (2002) and later Komatsu *et al.* (2005) and Nara *et al.* (2006) report that $\delta^{13}C$ of the major NMHCs emitted from biomass burning generally follows that of the fuel burnt, and the measurements did not reveal significant additional fractionations associated with the formation processes. Consequently, here (and further for the GFED data) the $^{13}C$ isotope fractionation escorting burning process is assumed to be negligible. On the contrary, the combustion conditions play a key role in formation of CO during the biomass burning: Normal ($+0.5\,‰$ to $+3.6\,‰$) and inverse ($-2.1\,‰$ to $-6.8\,‰$) $^{13}C$ fractionations were found to escort flaming and smouldering burning stages, respectively, with a further complex dependency on the burnt plant type (Kato *et al.*, 1999b). The average composition of CO is rather expected to be depleted with respect to the source fuel, since CO emission is expected to be favoured in the smouldering phase (Yokelson *et al.*, 1997). Unfortu-





nately, the representation of the combustion stages in the emission data is limited; hence, one can provide only a qualitative estimate of the isotope effect (depletion). The quantitative estimates of the contributions from various stages (like, for instance, in the modelling study by Soja *et al.*, 2004) could be improved with the use of the isotopic composition in this case. Conclusively, in contrast to the primary biomass burning sources, the emissions from the sector L43 induce a minor influence on the average CO emission signature, accounting for a total of 16.3 Tg(CO) per year (less than 3 % of the total anthro-

pogenic emission). In an analogous way, the waste treatment-related sources (sectors "W") are assigned to a slightly enriched (compared to the average fossil fuel carbon) composition of $-24\,‰$ using the ratio of the biological to fossil carbon for waste incineration from Johnke (2000). It is assumed that the waste treatment category refers to the waste incineration processes mainly.

[23] Table 2 lists the anthropogenic emissions and the compositions for the EDGAR database. The emissions for CO sum up

to almost 550 Tg yr$^{-1}$, while the overall influx for the other trace gases amounts to approximately 106 Tg(C) yr$^{-1}$. The mixing of the compositions of the main CO contributors (bio- and fossil fuel) in proportion of about 250:280, respectively, yields the average composition of $-26.15\,‰$. This value is apparently sensitive to the assumed biofuel $\delta^{13}$C signature. The influence of the biofuel sources is dominating for methanol, formaldehyde, formic acid, acetaldehyde and acetic acid, with values close to $-25\,‰$. Emitted alkanes and alkenes are enriched in $^{13}$C similar to CO, with an increasing influence of the fossil fuel input

towards the higher hydrocarbons. The spatial distribution of the $\delta^{13}$C of anthropogenically emitted CO is depicted in Fig. 2, with the panels referring to the specific emission altitudes, as described above. The two lowermost layers subsume the majority of the emission sectors, including the shipping and biofuel-related sources (equally distributed to the layers) and fossil fuel sources (falling mainly in the surface layer). The emission signatures reflect the dominant biofuel emissions in Africa, eastern Asia and Oceania (panel a). In the second emission layer (panel b) the agricultural waste burning and waste incinera-

tion sources are reflected together with the biofuel emission. The superincumbent layers include the mixture of industrial and power generation sectors, with the latter prevailing in the top two layers.

## 3.2    Biomass burning emissions

[24] The biomass burning emission data is prepared from the ORNL DAAC Global Fire Emission Database (GFED), version 2.1 inventory (Randerson *et al.*, 2007, http://daac.ornl.gov/VEGETATION/guides/global_fire_emissions_v2.1.html), an

updated and extended version of the initial GFED version 1 release (van der Werf *et al.*, 2006) used in the EVAL$_2$ setup (Pozzer *et al.*, 2009). In the current setup, monthly mean emission fields covering the period from 1997 to 2005 are used. The inventory includes emission fluxes for CO, NMHCs, nitrogen oxides (NO$_x$) and other species; in addition, the estimation of the C$_4$ plant carbon fraction of the burnt material is provided (Randerson *et al.*, 2005). The latter is used to assign the isotopic signatures to the emission fluxes, assuming negligible isotopic fractionation during the burning, as discussed above.

The resulting isotopologues fluxes are calculated as:




$$\begin{cases} \dfrac{^{13}C_F}{F} = 1 - f_{C_4} \; \dfrac{q \cdot R_{C_3}}{R_{C_3} + 1} + f_{C_4} \dfrac{q \cdot R_{C_4}}{R_{C_4} + 1} \\[3mm] \dfrac{^{12}C_F}{F} = 1 - f_{C_4} \; \dfrac{1 - q \cdot R_{C_3} + 1}{R_{C_3} + 1} + \\[3mm] \qquad + f_{C_4} \dfrac{1 - q \cdot R_{C_4} + 1}{R_{C_4} + 1} \end{cases} \tag{13}$$

The notation follows that from Eq. (1) and $f_{C_4}$ denotes the fraction of the burnt $C_4$ plant material, $F$ is the total emission flux. Ratios $R_{C_3}$ and $R_{C_4}$ refer to the $^{13}C$ isotope content associated with $C_3$ and $C_4$ plants, respectively; the corresponding isotopic signatures are discussed above. The emission is released into the second emission layer corresponding to 140 m height (see also Sect. 3.1).

[25] For the sake of comparison presented here, an averaged (ensemble mean) yearly biomass burning "climatology" was derived, referring to the 2000−2005 period of the original data. The "climatological" yearly average spatial distribution of a burnt $C_4$ biomass fraction and its translation into $\delta^{13}C$ values of the emission are presented in Fig. 3. The heaviest (*i.e.*, most enriched in $^{13}C$) composition of the emission is associated with the grassland and savannah burning regions, where the $C_4$ crops are most abundant.

[26] In Fig. 4 the temporal evolution of the hemisphere-integrated CO emission from biomass burning is presented. The markedly intensified emission rates in 1997−1998 are attributed to the increased wildfires due to the dry conditions and droughts induced by the enhanced atmospheric southern oscillation and El-Niño climate pattern (ENSO, Dube, 2009). This event is also notable (although less pronounced) for the years 2002−2003. Interestingly, ENSO activity is hardly reflected in the isotopic composition of the emission. However, the influence of the biomass source, especially important for its $^{13}C$ enriched composition in the tropics and southern hemisphere (SH), without doubt increases during El-Niño years. The variation of the flux $\delta^{13}C$ is twice as large in the northern hemisphere (NH) compared to that in the southern hemisphere. Such a difference arises from the large $C_3$ plant extent at the northern high latitudes and the pronounced seasonal fire cycle. The summer/fall extratropical fires in the NH occur predominantly in $C_3$ plant communities, mainly forests of an average −27‰ composition. In the winter time the (sub)tropical sources take over enriching the emission to the maximum of −19‰ due to the large $C_4$ plant fraction burnt in Africa and Asia. In the SH, the spatial diversity of the $C_3/C_4$ ratio is smaller over the smaller land extent, and the average signature varies around −24‰ within ±1‰ only.

[27] The annual average biomass burning emission rates for the relevant species are listed in Table 3. In contrast to CO, all NMHCs/VOCs emitted possess an equal isotopic composition because the fluxes for carbonaceous species are principally derived from the same burned carbon emission proxy (van der Werf *et al.*, 2006). In order to obtain the individual tracer emission, the proxy is scaled with the corresponding emission factor (conventions and values from Andreae and Merlet, 2001 are used), but the spatial distribution of the emission, hence $C_3/C_4$ carbon ratio, is the same. The hemispheric difference in $\delta^{13}C$ averages amounts to 0.4‰ with the heavier emission in the SH. For CO, a different proxy was used in GFED, which for the same burnt $C_4$ plant fraction results in a slightly heavier (+0.3‰ in $\delta^{13}C$) average composition. Notably, the





GFED v2.1 inventory provides the combustion completeness parameter, the estimate of the fraction of the actual fuel load combusted, which might to a certain degree reflect the burning stage conditions (*i.e.* flaming or smouldering phases). Unfortunately, the correspondence between these two parameters is not assessed to date; future applications of combustion completeness accounting for the kinetic isotope effects escorting biomass burning would be of great benefit.

## 3.3  Biogenic emissions

[28] The biogenic emissions represent the discharge of organic species into the atmosphere associated with biosphere activity, particularly oceanic, soil and plant emissions. The current biogenic emission setup in EVAL$_2$ follows Guenther *et al.* (1995) as described by Kerkweg *et al.* (2006), and comprises two parts for offline and online emissions, respectively (see the introduction in Sect. 2). The offline part is reassessed by Pozzer *et al.* (2007) and prescribes the emission for the large set of NMHCs/VOCs, excluding isoprene/monoterpenes emissions, which are calculated online. The data have a temporal resolu-

tion of one month, thus approximating the emission seasonal variation with no interannual variability. The emission is applied to the lowermost model layer. The CO emission comprises in-place oxidation of some (non-industrial) hydrocarbons not accounted for in the applied MECCA chemistry (*i.e.* higher alkenes (C>3), terpene products other than acetone, higher aldehydes) and some direct CO emissions by vegetation and decaying plant matter. The oceanic CO emission strengths (monthly zonal distribution) are taken from Bates *et al.* (1995). No biogenic emissions for formaldehyde (HCHO), acetalde-

hyde (CH$_3$CHO) and higher ketones (represented by methylethylketone (MEK) in MECCA) are included. The total annual emission strengths for CO and NMHCs/VOCs with the corresponding average compositions are listed in Table 4.

    [29] For the majority of the species, plant activity is the dominating biogenic emission. For a few species, *viz.* acetic acid (CH$_3$COOH), formic acid (HCOOH) and ethene (C$_2$H$_4$), the emission from the soils is estimated to be of comparable magnitude to the plants source (Kesselmeier and Staudt, 1999). Unfortunately, hardly any measurements or estimates of the isotop-

ic composition of the soil-emitted carbon of these VOCs are available. The composition of precursor soil organic matter is also not well known (Boutton, 1991). Regarding the example of methane, whose microbial production in soils is associated with large fractionations (Bréas *et al.*, 2001), soil emitted VOCs may constitute the source with the most uncertain signature. In case of CO, the aggregate of soil emissions is estimated to be negligibly small compared both to soil sink and overall CO turnover (Sanderson, 2002); even a radical change in its signature will be hardly reflected in the average $\delta^{13}$C(CO).

[30] A somewhat similar case arises with the oceanic emissions for which the strengths are debatable, and no isotopic signatures were estimated for NMHCs. Rudolph (1997) suggests the photochemical processing of dissolved organic carbon (DOC) to be the origin of C in the ocean-emitted NMHCs. Within the current setup an *a priori* signature of −20.5‰ representing the marine isotopic carbon content (Avery Jr *et al.*, 2006, lower limit) is assigned. This value is somewhat higher than −22‰ used for oceanic emissions by Stein and Rudolph (2007) in their modelling study on ethane isotopes. For CO,

heavier oceanic emissions of −13.5‰ are assumed, according to Manning *et al.* (1997). This value is based on the inverse modelling study and observations in the SH, where ocean input on CO is evidently significant. Quite contrary to this value,



Nakagawa *et al.* (2004) estimate the ocean emitted CO to possess a rather depleted composition of −40‰. This value appears to be still questionable, as the composition of the seawater-extracted CO was measured, the assumed precursor DOC composition was depleted (of average −31‰) and the sampling was done in a single, fairly non-remote location in waters with high microbial activity (thus likely escorted with significant kinetic fractionation during the production). Finally, Bergamaschi *et al.* (2000) estimate the composition of CO emitted from the oceans to be as high as +5.1‰ (scenario S2). Similar to biofuel-related sources, the oceanic CO is associated with a very uncertain isotopic composition. The change of this source signature from −13.5‰ to −40‰ will result in the decrease of the average biogenic emission signature by 3‰ with a corresponding 0.3‰ decrease in the overall CO surface emission composition.

### 3.3.1 Plant emissions

[31] For the plant biogenic emissions, a novel approach referring to the plant physiological properties is proposed here. In most previous (modelling) studies, the isotopic composition of the biogenically emitted tracers was based on the average global isotopic signature derived from the limited, often not consistent set of observations available. CO is a case in point here: The majority of the CO isotope modelling studies assume a $\delta^{13}C$ of CO emitted due the plant activity to be as low as −32.2‰, referring to the particular single estimate by Conny (1998). The latter was retrospectively derived from the observations at a rural US site (Stevens and Wagner, 1989), tolerating some important approximations, in particular (i) a two-component mixing model of the background and NMHC-only sources, (ii) constancy of the background composition throughout June to October, and (iii) neglecting the kinetic isotope fractionation caused by the CO sink. Whereas (i) is fairly applicable to the observations at a rural site, (ii) and (iii) rely on the five months constant background composition and neglect the variable input from the CO+OH reaction kinetic isotope effect (KIE). This is a too rough approximation, considering the intensive chemistry in the summer and characteristic CO lifetime shorter than a month. Indeed, the isotopic composition of background CO undergoes significant changes from spring to fall, and the competition of the CO+OH reaction KIE and the varying in-situ contribution from methane are the two non-negligible effects (Brenninkmeijer, 1993; Manning *et al.*, 1997; Röckmann *et al.*, 2002; Gromov *et al.*, 2010).

[32] Besides the temporal variation, the global average value does not represent the biogenic sources' variable spatial distribution, which is important, since biogenic CO is mainly a product of the rapid oxidation of NMHCs. The latter, in turn, are expected to acquire specific isotopic ratios being emitted from various plant species under different environmental conditions. The most studied compound in this respect is isoprene ($C_5H_8$), one of the major biogenically released VOCs. Sharkey *et al.* (1991) measured the carbon isotopic composition of the emitted isoprene and found it dependent on the composition of the recently fixed carbon, *i.e.* those from the air $CO_2$ incorporated in the plant material during the initial step of the photosynthetic cycle. The isotope effects related with the plant activity and plant-$CO_2$ exchanges are extensively studied (see, for instance, Dawson *et al.*, 2002). These usually operate with the isotope discrimination $\Delta$, a representative parameter describing the fractionation of the plant tissue relative to the atmospheric reservoir (Farquhar *et al.*, 1989):



$$\Delta = \frac{\delta_a - \delta_p}{1 + \delta_p}, \tag{14}$$

where $\delta_a$ and $\delta_p$ refer to the isotopic composition of the air $CO_2$ and plant tissues, respectively. In the form of Eq. (14), discrimination expresses the superposed effect of the various biological factors. The contribution of each of them, *e.g.* various plant metabolism pathways ($C_3$ or $C_4$, indices 3 and 4 indicate the number of carbons in the initial fixation product molecule), water availability for the plants (response to droughts), solar irradiance or various stress factors ought to be parameterised separately (Lloyd and Farquhar, 1994), hence it is a complex parameter. The largest effect on $\Delta$ is driven by the differences in the plant metabolism, the characteristic fixation mechanism of air $CO_2$ for the subsequent photosynthesis. The majority of the terrestrial plants incorporate the $C_3$ metabolism, when the fixation is escorted by the fractionation induced by RuBisCO (the specific enzyme used for the fixation in the so-called photosynthetic Calvin cycle). Accounting additionally for the other fractionations (*e.g.* diffusion of air $CO_2$ through the stomata, *etc.*), typical $\Delta$ values for $C_3$ plants span from 15 ‰ to 25 ‰. Note that discrimination is expressed on the positive scale. Assuming a certain $\delta_a$ (approximately −8 ‰ for air $CO_2$) and using Eq. (14), one derives the $C_3$ plant composition within the range of −32 ‰ to −23 ‰ in $\delta^{13}C$. $C_4$ plants employ other than RuBisCO enzymes; their efficiency is associated with lower $\Delta$ values of 2.5 ‰ to 5 ‰, corresponding to a −10 ‰ to −13 ‰ range of plant material $\delta^{13}C$. In addition to $C_3$ and $C_4$ plants, a minor fraction of terrestrial CAM (crassulacean acid metabolism) plants exists. CAM can be regarded as a temporal coupling of $C_3$ and $C_4$ metabolisms employed by the plant for optimised adaptation to arid conditions. Therefore, CAM plants are characterised with the wide range of discriminations from 2 ‰ to 22 ‰ (Griffiths, 1992), or −10 ‰ to −30 ‰ expressed in $\delta^{13}C$ of the plant tissue carbon. The specified plant biomass compositions result from the permanent isotopic equilibration with the atmospheric pool (*i.e.* $CO_2$) escorted by discrimination, thus the use of Eq. (14) is rational, when the long-term value of $\Delta$ is considered.

[33] In view of the correlation between the emitted species isotopic composition and the plant isotope discrimination, the latter is assumed here as a proxy for biogenic emission signatures in the current emission setup, rather than the global average signature. This approach, however, premises the following key assumptions:

– Few studies indicate that a moderate part (9 % to 28 %, Schnitzler *et al.*, 2004; Karl *et al.*, 2002) of the emitted isoprene may be issued from a separate carbon source of the plant. Its composition may differ from that expected from $\Delta$, the photosynthetically fixed carbon. Moreover, neither the isotopic composition of the suggested alternative sources was deduced, nor the fractionations associated with their incorporation in the emission product. Affek and Yakir (2003) overcame this issue showing that the long-term value of $\Delta$ may be used as a proxy for the average bulk leaf biomass, thus concluding the depletion of the emitted isoprene in relation to it. It is important to note that the contribution of alternative sources becomes larger as the plant is put under stress (*e.g.*, experiments of Schnitzler *et al.* (2004) were partly carried in $CO_2$-free air). For natural conditions, the proportion of the non-photosynthetically fixed carbon is likely to be smaller.

– The abovementioned studies have analysed exclusively isoprene; no comparable measurements were performed regarding the other species. Nevertheless, there are isotopic compositions of biogenically emitted NMHCs/VOCs reported rela-





tive to the plant bulk leaf composition (Rudolph *et al.*, 2003; Sharkey *et al.*, 1991; Conny and Currie, 1996), as well as few measurements of the plant-emitted VOCs whose $\delta^{13}C$ is found comparable to that of the expected bulk composition (Giebel *et al.*, 2010). Thus, it is practicable to derive the emission signatures from the measured depletions of the trace gas composition relative to that of the plant leaf. It is tolerable under the assumption that the latter is determined by the long-term value of $\Delta$ yielding from the specific plant metabolism and diffusion/equilibrium effects of the $CO_2$ photosynthetic fixation and respiration.

[34] For constructing the emission signatures, the estimated global distribution of the leaf discrimination is taken from Scholze *et al.* (2008). They use a dynamic global vegetation model extended with the terrestrial isotopic carbon module. The parameterisation of the leaf carbon discrimination is based on the framework by Lloyd and Farquhar (1994) neglecting poorly understood fractionations in several processes involved in the photorespiration. The vegetation dynamics model accounts for the plant and soil carbon reservoirs and a numerous set of parameters including the vegetation composition, its productivity, fire disturbance, water availability and land use schemes, as well as climate forcing (monthly temperature, precipitation and cloud cover fields). For the detailed model description, the reader is referred to Scholze *et al.* (2003) and the abovementioned references. The simulated leaf discrimination for the year 1995 from the ISOLUCP experiment (depicted in Fig. 5, left panel) is adopted here. The characteristic variability of the global leaf discrimination magnitude is on the order of decades, thus the data referring to 1995 is reckoned to be consistent for the studied year 2000. The bulk leaf composition $\delta_p$ is calculated straight from the isotope discrimination defined in Eq. (14), for which the isotopic composition of $CO_2$, namely $\delta_a$, is required. For the period of 1997−2005 (corresponding biomass burning data in the current setup), the estimate of the surface $CO_2$ isotopic composition from the GLOBALVIEW project (GLOBALVIEW-CO2C13, 2009) is taken. These data comprise latitudinal weekly averages (shown in Fig. 5, right panel), thus the latitudinal mean of the $\delta^{13}C(CO_2)$ went into the calculations. Except for isoprene, the fractionations accompanying the emissions are considered to be negligibly small, as no significant deviation (within measurement standard deviation of 1‰) from the source plant material for the selected NMHCs was reported (Conny and Currie, 1996; Guo *et al.*, 2009). For the fractionation escorting isoprene emission, the lower limit of 4‰ depletion relative to the bulk leaf composition from Affek and Yakir (2003) is taken.

[35] The biogenic emission strengths and resulting isotopic signatures (average values for the year 2000) are listed in Table 4. The largest offline emissions pertain to CO and methanol. The final signatures reflect the proportion of the land (average −25.7‰) and oceanic sources. The average composition of the CO emission of −24.2‰ is perceptibly $^{13}C$-enriched compared to the previously assumed −32.2‰ (Conny, 1998), results in an effective increase of about +0.8‰ in the overall surface emission $\delta^{13}C$. The major part of the emissions is placed in the tropics, with the summer-triggered large emission in the NH. An example for CO is sketched in Fig. 6. The largest influx is associated with the areas of rather depleted sources. The land sources are weaker than the oceanic sources in NH winter, which is markedly reflected in the isotopic composition of CO emissions. Based on the same proxy, the emission signature dynamics is similar for the other species.



[36] The isoprene emission, in turn, is calculated on-line, utilising model parameters obtained during the calculation. The emission parameterisation is described by Ganzeveld *et al.* (2002) and implemented for EMAC in the ONEMIS (formerly ONLEM) sub-model (Kerkweg *et al.*, 2006). The key variables for the $C_5H_8$ emission are the temperature and radiative balance over the canopy (both are provided by the base model) and the vegetation foliar density (prescribed). The isoprene influx is calculated every model time step from the abovementioned variables. To account for the isotopic $C_5H_8$ emission, the necessary extension to ONEMIS was implemented. The influxes of the $^{12}C/^{13}C$ isotopologues are calculated from the original isoprene emission flux and either simulated or prescribed average $CO_2$ isotopic composition. The leaf discrimination distribution is imported as a parameter (similar to the other prescribed data fields). The overall $C_5H_8$ emission amounts to approximately $350-380$ Tg yr$^{-1}$ with the corresponding average $^{13}C$ signature within the range of $-28.6‰$ to $-27.2‰$ depending on the proportional contributions from the source regions. As indirect (in-situ oxidation) source of CO, isoprene dominates over the sum of all remaining VOCs accounted for in the setup.

### 3.4 Final composition of the surface sources

[37] Table 5 lists the annually integrated trace gases emissions from the surface in the reference emission setup of this study. For the carbonaceous species, stable carbon isotopic compositions resulting from the superposition of the various emission types are given; values refer to the year 2000. The inter-annual variation for $1997-2005$ of the average $\delta^{13}C$ signature of emitted CO is less than $0.5‰$ yr$^{-1}$ resulting from the variability of $\pm0.6‰$ yr$^{-1}$ in the biomass-burned carbon and a negative trend in the $CO_2$ composition in the last decades ($-0.02‰$ to $-0.03‰$ yr$^{-1}$ due to the input of fossil fuel-derived carbon into the atmosphere, Yakir, 2011) propagating into the biogenic emissions.

[38] The spatial distribution and annual dynamics of the surface CO emission is presented in Fig. 7. The largest emission is situated in the tropics, particularly in Africa and Asia and attributed to the biomass burning season in July-September in the SH, African fires in December and high-latitude fires in Eurasia and Northern America from May to September. A comparable emission proportion is made up by the anthropogenic sources, which have no distinct seasonality and are present in the NH high latitudes; these are mostly transportation and industry (*i.e.*, fossil fuel related sources). The relative dynamics of the isotopic composition is weaker than that of the corresponding flux magnitudes, indicating that the dominant sources are close to the average $-25‰$ to $-27‰$ of terrestrial carbon, with the exception of the North African and Australian fires, when a significant proportion of $C_4$ plants is being burnt. The largest portion of $^{13}C$-enriched CO enters the atmosphere from December to March from the African equatorial fires. Interestingly, mixing of the fossil fuel-derived CO from ships and the heavier oceanic CO emissions highlights the most navigated ship tracks in the $\delta^{13}C(CO)$ map, where the strengths of these sources become comparable.

[39] The average compositions of the majority of NMHCs/VOCs fall in the range of $-26‰$ to $-24‰$ with the exception of isoprene, propane and butane (Fig. 8). For the latter two, the emission is coming predominantly from anthropogenic sources, which are close to $-27‰$. The isoprene composition reflects an assumed $4‰$ depletion from the average terrestrial carbon



composition. The annual emission dynamics for NMHCs/VOCs generally follows the proportion of the sources, *e.g.* variations for $CH_3OH$ and $CH_3COCH_3$ are mainly driven by the biogenic emission. The particular source dynamics for various NMHCs/VOCs resemble each other being derived from the same proxies (*e.g.* burnt carbon in GFED). The uncertainties associated with emission fluxes and corresponding isotope signatures are discussed below in Sect. 3.6.

### 3.5 Pseudo-emission data

[40] For the few long-lived tracers in the current setup the pseudo-emission approach is applied by performing the relaxation of the selected species mixing ratios towards the lower boundary conditions (see also Sect. 2 above). The relaxation is handled by the TNUDGE submodel (Kerkweg *et al.*, 2006) and applied at every model time step with typical relaxation times of 3 h for the less reactive compounds (*e.g.* $CH_4$, $CO_2$, $N_2O$, *etc.*) . The nudging fields are based on the observed mixing ratios from the AGAGE database (Prinn *et al.*, 2000). Amongst the tracers undergoing nudging, $CH_4$, $CH_3CCl_3$, $CCl_4$, $CH_3Cl$, and $CO_2$ are isotopically separated. For $CO_2$, the time series of the zonally averaged composition from the GLOBALVIEW-CO2C13 database (described above in Sect. 3.3.1, see also Fig. 5) was superimposed on the regular $CO_2$ nudging fields from the EVAL$_2$ setup.

[41] Methane ($CH_4$) is the major atmospheric in-situ source of CO and other reactive carbonaceous species participating in the $CH_4 \rightarrow CO$ oxidation chain. Tropospheric $CH_4$ possesses a markedly [13]C-depleted composition, particularly due to the large contribution of the sources associated with the biogenic activity that prefers emitting isotopically light methane (see Bréas *et al.*, 2001 and references therein). The average tropospheric $\delta^{13}C(CH_4)$ value of $-47.3\,‰$ (corresponding the year 2000) ensues from the composition of the surface sources (estimated equilibrated average of $-51.2\,‰$) and atmospheric oxidation KIEs, of which the reaction with OH ($+3.9\,‰$) is the dominant in the troposphere (Saueressig *et al.*, 2001). Since methane is largely abundant and long-lived, its signature shows a low variability on top of a weak long term trend (about $+0.3\,‰$ per decade around the year 2000, Lassey *et al.*, 2000) due to the input of the industrial fossil carbon, and little spatial and temporal variability. Quay *et al.* (1999) estimated the hemispheric gradient (averages of $-47.2\,‰$ versus $-47.4\,‰$ for the SH and NH, respectively) and the monthly variation of $\delta^{13}C(CH_4)$ to be both on the order of $\pm0.2\,‰$. That is negligible in view of $\pm3\,‰$ variations in tropospheric $\delta^{13}C$ of CO and its large surface sources. Therefore, the constant value of $-47.2\,‰$ is applied to isotopically separate the original nudging fields of $CH_4$ in the current setup.

[42] Among the chlorinated carbons, the only source of isotopic carbon accounted for in the employed chemical mechanism of MECCA (as of EVAL$_2$ setup) is the photolysis of chloromethane (decomposing to $CH_3O_2$). The remaining chlorinated carbons contribute only as the in-situ sources of chlorine, thus their composition is omitted here. The main sources of chloromethane in the atmosphere are to date not clearly identified (Keppler *et al.*, 2005), the estimate of the average global isotopic atmospheric composition is $\delta^{13}C(CH_3Cl) = -32.6\,‰$ (Thompson *et al.*, 2002). This value is used for the pseudo-emission of chloromethane. The contribution of this source to the carbon pool in the atmosphere is low. The estimates of the primary



CH$_3$Cl sink through the reaction with OH give a global average of 3.37 Tg(CH$_3$Cl) yr$^{-1}$ equivalent to 0.8 Tg(C) yr$^{-1}$ in the oxidised products (methyl peroxy radical).

## 3.6 Uncertainties

[43] In order to calculate the overall emission uncertainties in this study, we account for uncertainties associated with every emission source and its isotope signature, following the methodology described above (Sect. 2.2). The emission magnitudes and uncertainties are expressed in equivalent carbon units to avoid improper counting when isotope ratios are considered. Table 6 lists the uncertainties associated with every emission category/sector. For the fluxes, the so-called uncertainty factors (UF) are quoted, which are commonly reported in emission estimates and refer to a given confidence interval (CI) of emission flux (or typically underlying emission factor) with a given uncertainty probability density distribution (UPDD). For example, the UF of 1.5 may imply that the 95 % CI of uncertainty spans from $F/1.5$ to $1.5 \cdot F$, or, in percent, from about $-33$ % $F$ to $+50$ % $F$, describing a log-normal UPDD around the median value of $F$. Exceptionally, the UFs reported for the EDGAR inventory (see Olivier *et al.*, 1999, Table 8) indicate the equivalent span (*i.e.*, Gaussian or any symmetric UPDD) range derived from the largest (*i.e.* upper end) value, that is for the above example would be $\pm 50$ % $F$ around $F$. Such treatment is used in our analysis here (including reporting with the "±" notation) too, that is, selecting the largest (forward) uncertainty $\langle F \rangle$ using the relation

$$\frac{\langle F \rangle}{F} = u_F - 1 \ ,$$

where $u_F$ is the uncertainty factor. Diversely, signatures' uncertainties are reported plainly in $\delta$-units assuming normal (Gaussian) UPDD, as the isotopic ratios do not depend on the flux magnitudes.

[44] The uncertainties for some of the signatures have to be derived additionally, referring to the assumptions they are based on. For the composites of the different plant material, the uncertainties of C$_3$ and C$_4$ signatures contribute to the final uncertainty, similarly using Eq. (11) and substituting for $\langle F_s \rangle$ the respective fractions. The uncertainties of C$_3$ and C$_4$ plant matter signatures itself are inferred as two standard deviations of the signature distributions (assumed normal) based on the histogram data of the measured terrestrial compositions (Cerling *et al.*, 1999; Tipple and Pagani, 2007). The isotopic composition variability in C$_3$ plants is much larger than that of C$_4$, which is reflected in the resulting uncertainties of $\langle \delta^{13}C(C_3) \rangle = 5.7$ ‰ and $\langle \delta^{13}C(C_4) \rangle = 2.5$ ‰, respectively. This means that if, for instance, the plant is considered to be of the C$_3$ kind, its composition is likely to be found within the range of $\delta^{13}C(C_3) = -(27 \pm 2.5)$ ‰. From the "emission guess" point of view, the uncertainty defines the degree of error introduced by assuming all C$_3$ plants to have the composition of the C$_3$ distribution mode of $-27$‰. The errors associated with the plant compositions are the largest in this setup and they propagate to the final uncertainty mainly via the biofuel category. Interestingly, if one assumes that biofuel plant material comes predominantly from C$_4$ plants (*e.g.*, ethanol or biodiesel, see Sect. 3.2), it significantly decreases the overall uncertainty estimate.





[45] An additional calculation is required for those biogenic emissions originating from plants, whose signatures are derived from the leaf discrimination $\Delta$ and air $CO_2$ composition (see Eq. (14)). The uncertainty of the latter is on the order of 0.01 ‰ according to the GLOBALVIEW-CO2C13 dataset (see http://www.esrl.noaa.gov/gmd/ccgg/globalview/gv_integration.html and references therein; here twice that value is assumed). The errors in $\Delta$ are as large as 2 ‰, taking one standard deviation
of the comparison of the simulated and measured characteristic discriminations for various plant functional types (Scholze *et al.*, 2008). The resulting propagated uncertainty amounts to $\langle \delta_p \rangle = 1.9$ ‰ (at the average global discrimination of $\Delta = 17$ ‰ and $\delta^{13}C(CO_2) = -8$ ‰) and accounts for all plant emissions, whose UFs of the magnitude of 3 are the largest (Guenther *et al.*, 1995). The biomass burning signatures uncertainties are set to 2 ‰ referring to the upper limit of errors in atmospheric $\delta^{13}C$ used to validate the $C_3/C_4$ burnt vegetation distribution incorporated in the GFED v2.1 inventory
(Still *et al.*, 2003). The UFs for biomass burning emissions are derived from the uncertainties on the estimates for global CO and carbon release in fires by Arellano *et al.* (2006) for the April 2000 to March 2001 period obtained using the GFED data (van der Werf *et al.*, 2006).

[46] Employing the methodology described in Sect. 2.2, we derive the resulting overall (combined) uncertainties (listed in Table 5). Essentially high uncertainties are associated with isoprene and plant-dominated emissions of methanol ($CH_3OH$), ace-
590 tone ($CH_3COCH_3$), dimethyl sulphide (DMS) and formic acid (HCOOH). The errors are lower (UFs of 1.5−2) for the species predominantly emitted from the fossil anthropogenic sources. Final uncertainties associated with the isotopic signatures are typically around 1 ‰, with the biofuel source having a large contribution of (0.3−0.4) ‰. The terrestrial emissions are least uncertain resulting from the lower error in leaf carbon discrimination compared to the uncertainties from $C_3/C_4$ plant composites.

[47] Despite the large share of the biofuel sector emissions, the uncertainty of the CO $\delta^{13}C$ signature is 0.7 ‰ due to the compensating input from the fossil fuel sector with a signature of a higher certainty (0.3 ‰). The final emission strength is defined within ±17 %, yet a rather large value. Reckoning the surface sources of about 1100 Tg yr$^{-1}$ in the global turnover of CO of above 2600 Tg yr$^{-1}$ (see the estimates in the following section), the emission uncertainties are expected to propagate in the model result errors with at most ±30 % in CO mixing ratios and ±1.3 ‰ in $\delta^{13}C(CO)$, respectively. To estimate the uncer-
tainties associated with the in-situ produced CO, the emission/isotope signature uncertainties of the respective NMHC/VOC sources should be used as the proxies accordingly.

## 4 Discussion

### 4.1 $^{13}CO/^{12}CO$ emissions

[48] Table 7 lists our resulting $^{13}C/^{12}C$-resolved CO emission inventory compared with the estimates available from previous
studies. Notably, the bottom-up estimates (including the *a priori* setups for the inverse modelling studies) integrate more $^{13}C$-depleted fluxes and vary less significantly between different studies, *i.e.* within −35 ‰ to −33 ‰ in $\delta^{13}C$. The earliest top-





down estimate of $-30.3\,‰$ given by Stevens and Wagner (1989) (hereinafter denoted "SW89") is based on the average of the atmospheric $\delta^{13}C(CO)$ observed by that time, corrected for the average tropospheric $^{13}CO$ enrichment (reckoned to be $+3\,‰$) caused by the KIE escorting CO removal by OH. Similar to SW89, the *a posteriori* estimates from the inverse modelling studies favour the overall CO source $\delta^{13}C$ of $-31.1\,‰$ to $-30.5\,‰$ resulting from the larger $^{13}C$-enriched surface influx and reduced methane oxidation source shares. The difference between the bottom-up and top-down estimates of the primary sources is $3-4\,‰$, which, if one assumes the CO yield from $CH_4$ oxidation being nearly unity, causes an even larger disparity in the estimates of the average $\delta^{13}C$ of the non-$CH_4$ CO sources. Thus, from Manning *et al.* (1997) ("M97") and Bergamaschi *et al.* (2000) ("B00") these should be $-21.3\,‰$, whereas for the other studies the non-methane CO source signature is much lower, *e.g.* $-26.1\,‰$ in Emmons *et al.* (2004) ("E04") and $-25.2\,‰$ (this study, EVAL$_2$). From the CO budget considerations of Brenninkmeijer *et al.* (1999) ("B99") one derives similarly $^{13}C$-depleted source composition, when superimposing the respective $\delta^{13}C$ values from the literature on their reported emission strengths.

[49] Fig. 9 (right panel) details the global CO source by category from the previous and current isotope-enabled studies. Neither bottom-up nor top-down estimates show correlated tendencies, suggesting the overall CO budget being uncertain within at least $\pm200\,Tg(CO)\,yr^{-1}$. A similar estimate of about $2700\pm280\,Tg(CO)\,yr^{-1}$ one infers from the results of the ensemble of the inverse modelling approaches summarised by Duncan *et al.* (2007), narrowed down to $2500\pm185\,Tg(CO)\,yr^{-1}$ for the year 2000 (see refs. therein; quoted is the ensemble average $\pm1$ standard deviation, respectively). The large variation of $2500-2900\,Tg(CO)\,yr^{-1}$ of these estimates (quoted range refers to the year 2000 or to the interannual averages conferred by the studies regarded) is generally attributed to the differences in the implementation of inverted surface emission strengths. Regarding the variation range of individual CO sources between the studies, the largest spread of around $280\,Tg(CO)\,yr^{-1}$ (or equivalent $50\,\%$ of its average value) is attributed to the biomass burning (BB) source. The most ambiguous biogenic source (including oceanic emission) is varying within around $70\,\%$ of its average, or $90\,Tg(CO)\,yr^{-1}$, but is nonetheless least influential in the aggregate emission composition. The moderately uncertain fossil fuel/biofuel (FF/BF) and VOCs oxidation sources range within about $25\,\%$ and $30\,\%$ (170 and $150\,Tg(CO)\,yr^{-1}$), respectively. Disregarding the rather low estimates of M97 and B00, the methane source of CO appears the most certain one ranging only within $15\,\%$, or roughly $110\,Tg(CO)\,yr^{-1}$ around its average value.

[50] Amongst the studies regarded here, the *a priori* and bottom up derived sources sum up to about $2900\,Tg(CO)\,yr^{-1}$, *i.e.* lie at the upper end of the range quoted above. The *a posteriori* sources in M97 are generally reduced at the expense of the smaller $CH_4$ source. In contrast to it, B00 decrease the methane-derived CO less and compensate it by other sources, thus keeping the final emission strengths close to the initial guess. Noteworthy, these two studies also infer the largest BB emission sources exceeding the inter-study average by a factor of $^2/_3$ and $^1/_3$, respectively. A significantly lower CO budget in M97 is most probably the drawback of using the fairly limited observational data from the extra-tropical SH, where the inversion results are less sensitive to the NH sources, or their underestimation. Comparably low CO emissions for EMAC are derived here, which, when applied, are likely to result in systematically low simulated NH high-latitude CO mixing ratios,



particularly in winter. A similar feature was observed in the previous studies with EMAC (Pozzer *et al.*, 2007, their setup is being closely followed here, see Sect. 2), as well as in other models/inventories employed (*e.g.*, B00 and E04, see also Stein *et al.*, 2014, and refs. therein). Stein *et al.* (2014) show that a more detailed representation of the strength and seasonality of CO dry deposition fluxes and traffic emissions in Europe and North America leads to more adequately reproduced NH CO mixing ratios. Noteworthy, their hypothesis that the missing traffic CO is due to emission inventories not accounting for

cold-start engine conditions should be verifiable through $^{18}O/^{16}O$, but unfortunately not by $^{13}C/^{12}C$ ratios of emitted CO (see Kato *et al.* (1999a), also Sect. 3.1). Nevertheless, it is clear that strengths and spatial distribution of the missing CO sources shall receive a more thorough quantification through the isotope-resolved inventories, which we undertake in subsequent studies.

[51] In addition to the comparison of the CO sources strengths, left panel in Fig. 9 elucidates individual contributions of every

source term to the $\delta^{13}C$ of total emitted CO in the isotope-inclusive budget inquiries. The source terms (bars) are calculated as the products ($f_s \cdot \delta_s$), where $f_s$ is the fractional contribution and $\delta_s$ is the $\delta^{13}C$ of a particular CO source, respectively. This way one grasps the integration of individual inputs enriching/depleting the final composition (with respect to the reference ratio of 0‰), which also highlights the inter-study variation of each source' input. Because the majority of the CO sources is depleted, the calculated contributions are always negative, with an exception of the minute term of +0.1‰ in B00 from the

oceanic source with a corresponding $\delta_s = +5.1$‰ (added up to the biogenic category). Due to the appreciably $^{13}C$-depleted composition of methane (−51.2‰), the overall composition is highly sensitive to the $CH_4$ source input, with clearly smaller contributions in M97 and B00. In contrast, the variation in the total surface source input to $\delta^{13}C$ is rather low, as opposed to the variation in respective fluxes.

[52] Coherent adjustments to the source composition in the *a posteriori* estimates are given by the inverse studies, however

they remain within the uncertainty ranges of the *a priori* guesses (note that these are based on different isotope signatures as well, not listed in Table 7). Despite the improved uncertainties for almost each individual source category, the combined (either surface or total) *a posteriori* source estimates' uncertainties are essentially larger than those of the prior guesses, owing to the correlated nature of the inverted components (see Sect. 2.2 for elucidation). Thus, posterior combined uncertainties increase by a factor of 1.3−1.7 (fluxes) and 2.4−3.1 (flux $\delta^{13}C$ values) with respect to those of the independent priors, respec-

tively. An exception is the reduction of uncertainty in the overall surface CO flux (factor 0.8) but not of its $\delta^{13}C$ value (increase, factor 1.2) in B00, which, however, does not reduce the final overall uncertainty.

[53] Furthermore, on a global scale the posterior repartitioning of the non-methane sources is virtually ineffective in M97: An increase of +2.7‰ in $\delta_s$ of the VOC oxidation source counterbalances the sufficiently larger BB source in the optimised emissions, hence the increase in tropospheric $\delta^{13}C(CO)$ is merely promoted by adjusting the $CH_4$ source. The reduction of

the methane component in B00 is less marginal, whilst the non-methane sources also deplete the final $\delta^{13}C(CO)$ less, being enriched by a similar adjustment of the VOC signature by +2.5‰. Despite the fact that the $CH_4$ source strength inferred by B00 is comparable to the majority of the estimates presented in Fig. 9 (right panel), its relative contribution to the overall CO





is diminished by a larger fraction of the other sources, which is a direct consequence of the reduced CO yield (0.86) from CH$_4$. The remaining studies suggest almost complete conversion of the CH$_4$+OH source to CO, and by this confine the overall source δ$^{13}$C to the −35 ‰ to −33 ‰ range. The results of the inversion studies (including the top-down estimate of SW89) importantly retain the *expected* tropospheric average of above −28 ‰ "assimilated" to a considerable extent from the observational data at the surface. Regarding the bottom-up estimates, it becomes clear that the CO+OH sink fractionation, when assumed to be about +3 ‰, is capable of bringing the tropospheric δ$^{13}$C(CO) value at most to −30.5 ‰, that is a perceptibly underestimated $^{13}$CO/$^{12}$CO tropospheric ratio.

## 4.2 $^{13}$C/$^{12}$C emission ratios of NMHCs/VOCs

[54] Only one $^{13}$C-inclusive global-scale emission estimate for ethane is available to date for comparison with the NMHC/VOC emissions derived here. Using two 3D chemical transport models (CTM), Stein and Rudolph (2007) (hereinafter "SR07") evaluate two emission sets based on the GEIA/EDGAR inventories (detailed in Sect. 2), which differ in inclusion of the biofuel, biogenic and oceanic sources. Integrating the same literature sources (listed in Sect. 3), the authors use slightly different assumptions on the isotope composition of emitted C$_2$H$_6$, namely δ$^{13}$C signatures of C$_4$ plant carbon of −13 ‰, fossil-fuel carbon of −26 ‰ and gas production and transmission of −32 ‰, respectively. Furthermore, anthropogenic emission fluxes in SR07 are based on the previous version (2.0) of the EDGAR inventory. Being optimised in simulations with CTMs, emissions in SR07 offer more independent comparison against the current results based on the newer version (3.2) of EDGAR (see Sect. 3.1).

[55] Both SR07 estimates of C$_2$H$_6$ emission fluxes are lower than, but within the uncertainty range of, the estimate reckoned here, *i.e.* 8.2 in MOZART CTM emissions ("MOZ") and 9.57 in GISS CTM emissions ("GISS") compared to 12.48±5.49 Tg(C$_2$H$_2$) yr$^{-1}$ in EMAC, respectively. The δ$^{13}$C of total emitted ethane (−28.5 ‰) in MOZ is virtually identical to the value derived here (see Table 5), however it is composed of very different relative inputs (that is, the $f_s \cdot \delta_s$ terms, see previous section). Their shares (FF+BF : BB : biogenic) are lighter in the anthropogenic component in MOZ (−13.8 ‰ : −9.6 ‰ : −2.4 ‰) *vs.* that in EMAC (−19.6 ‰ : −5.3 ‰ : −0.9 ‰, respectively). Projecting the δ$^{13}$C signatures of MOZ onto the GISS fluxes yields slightly lower overall emission δ$^{13}$C of −26.6 ‰ (−19.8 ‰ : −6.8 ‰ : n/a), which is still on the lower end of −(25.9±0.8 ‰) obtained in EMAC. A similar projection of the emission δ$^{13}$C signatures used by SR07 onto the emission fluxes in EMAC, and *vice versa*, yields the large span of the overall emission δ$^{13}$C value of −(18.6−22.4) ‰, which suggests that the $^{13}$C-resolved C$_2$H$_6$ emission inventories should be rather sensitive to the ratio of anthropogenic and biogenic inputs. In this respect, the results obtained here for EMAC reconcile both the underestimated anthropogenic sources highlighted by SR07 and their (top-down) estimate of the global ethane δ$^{13}$C signature.

[56] SR07 do not provide a detailed uncertainty analysis for their emission estimates. Nonetheless, we attempt to derive these by applying the analysis and uncertainty factors reckoned for EMAC here (see Sect. 3.6, also Table 6), since similar emission categories and same literature sources are used. Thus derived global emission flux uncertainties in SR07 are of ±29 % and



±32 % in MOZ and GISS, respectively, and are noticeably lower than ±44 % in EMAC, mostly owing to the different treat-
ment of the BF sources (these are assumed by SR07 known with greater certainty, *i.e.* that of the FF sources). In contrast, the
overall $\delta^{13}$C signature uncertainties are only slightly improved w.r.t. to that in EMAC, *viz.* to ±0.7 ‰ and ±0.6 ‰ in MOZ
and GISS, respectively. We therefore may conclude that all three estimates considered here agree in strength and isotope
ratio of the global ethane emission flux.

## 5 Concluding remarks

[57] In this study, we attempt to deliver a comprehensive to date review on the $^{13}$C/$^{12}$C ratios of emission sources of atmos-
pheric CO and other reactive carbonaceous compounds. As a consistent starting point for the isotope extension, we choose
the evaluated emission setup of the EMAC model (EVAL$_2$, see Sect. 2). The latter does not employ the most recent versions
of some inventories (*e.g.*, EDGAR), however, we believe the information on proxies and the uncertainty analysis offered
here should suffice and enable one to perform a complete isotope extension of any desired up-to-date inventory in a fashion
similar to that presented here.

[58] Compiling the isotope-inclusive emission inventory immediately highlights several peculiarities of the $^{13}$CO budget in
comparison with previous studies. First, we corroborate that the bottom-up and top-down estimates disagree on the overall
surface-emitted CO isotope signature, with the top down approaches reckoning it to be (2−3) ‰ heavier in $\delta^{13}$C. This dis-
crepancy is larger than the associated uncertainties in all studies regarded here (an exception is the *a posteriori* estimate of
M97) and calls further for clarification. Second, we note that our estimate has a substantially lower uncertainty (±0.7 ‰) as-
sociated with the total surface emission term. Furthermore, accurate use of probabilistic calculus renders the inverse model-
ling studies delivering *a posteriori* global estimates that are generally less certain than their *a priori* guesses. This may leave
bottom-up approaches favourable, as an increase in boundary condition data fed into inverse models does not necessarily
reduce posterior uncertainties to adequate levels (*cf.* uncertainties in M97 and B00 with the latter utilizing a substantially
larger set of observational data). Third, isotope mass-balancing of the CO sources is very sensitive to the input of $^{13}$C-
depleted carbon from the CH$_4$ oxidation source (*cf.* Fig. 9 and Table 7), with the key question being the tropospheric yield of
CO from methane oxidation. The latter is another aspect (mutual to the one outlined above) of disagreement between the
bottom-up and top-down approaches, which is not reconciled yet. Perhaps, a hybrid iterative approach consisting of inverse
modelling steps (performing optimisation of the emission fluxes only), followed by forward modelling steps (applying less
uncertain bottom-up isotope signatures), could offer an efficient solution to this problem.

[59] At last, the comparison of our results with the study by SR07 on isotope-resolved ethane emissions evidences that isotope
ratio information may bring deeper insight into studies dealing with NMHCs/VOCs as well, even at the stage of compiling
the emission inventories, *e.g.* comparing their versions. We therefore hope that current results will bolster the community for
further efforts in this yet little explored area of atmospheric isotope composition modelling field.



***Acknowledgements.*** Authors are grateful to Alan Goldstein (UC Berkeley), Elena Popa (IMAU Utrecht) and Taku Umezawa (NIES Tsukuba) for fruitful discussions on CO and trace gas emissions and their isotope composition particularities. Andrea Pozzer (MPI-C Mainz) is acknowledged for the great help with the biogenic emissions in EMAC.

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



# Figures

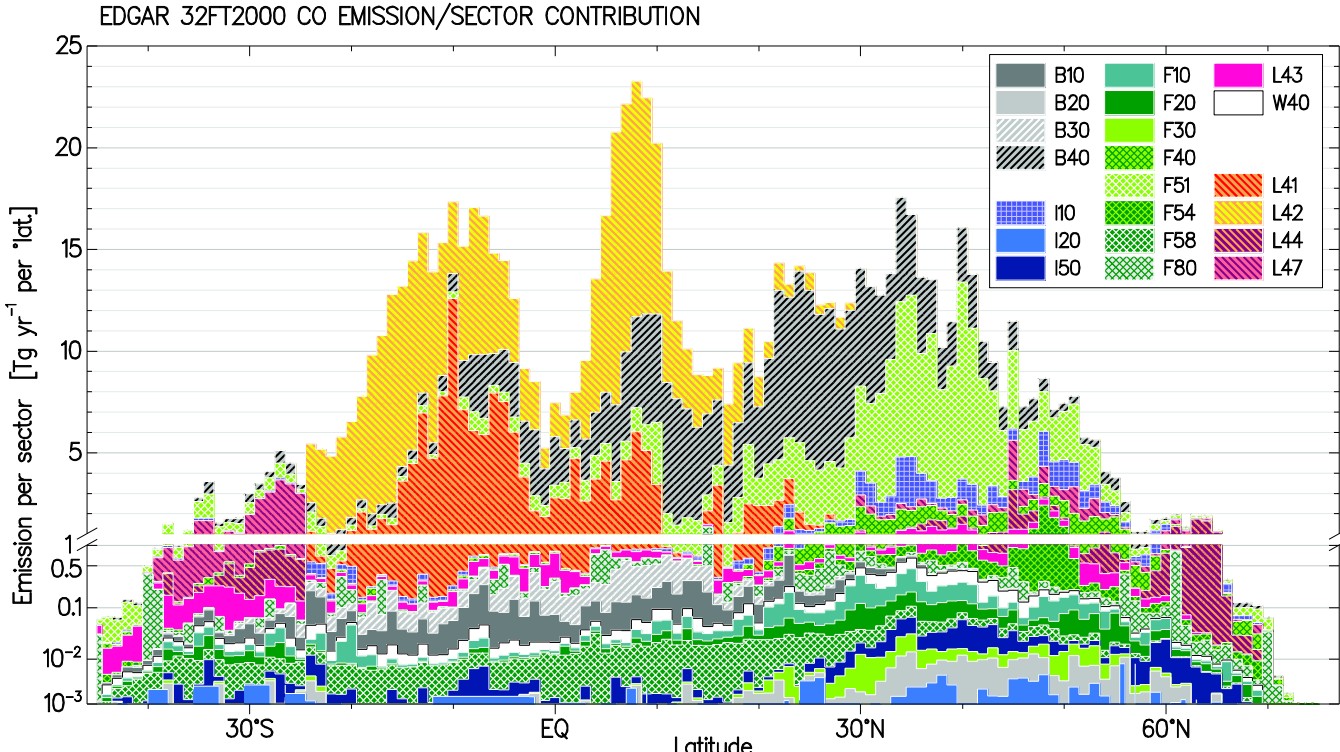

Fig. 1 Relative contributions of emission sectors to the overall emission of CO in the EDGAR inventory. Values are given in Tg(CO) yr$^{-1}$ per degree latitude. Note: The original EDGAR biomass-burning sectors L41, L42, L44 and L47 are presented here for comparison only. They are being substituted (see text) by the GFED inventory. Mind the change in ordinate axis scale at the value of unity.



Fig. 2 Stable carbon isotope composition of CO emitted from anthropogenic sources compiled on the basis of the EDGAR FT2000 inventory. Panels (a)-(f) refer to the specific emission heights of 45, 140, 240, 400, 600 and 800 m, respectively (see text for details).



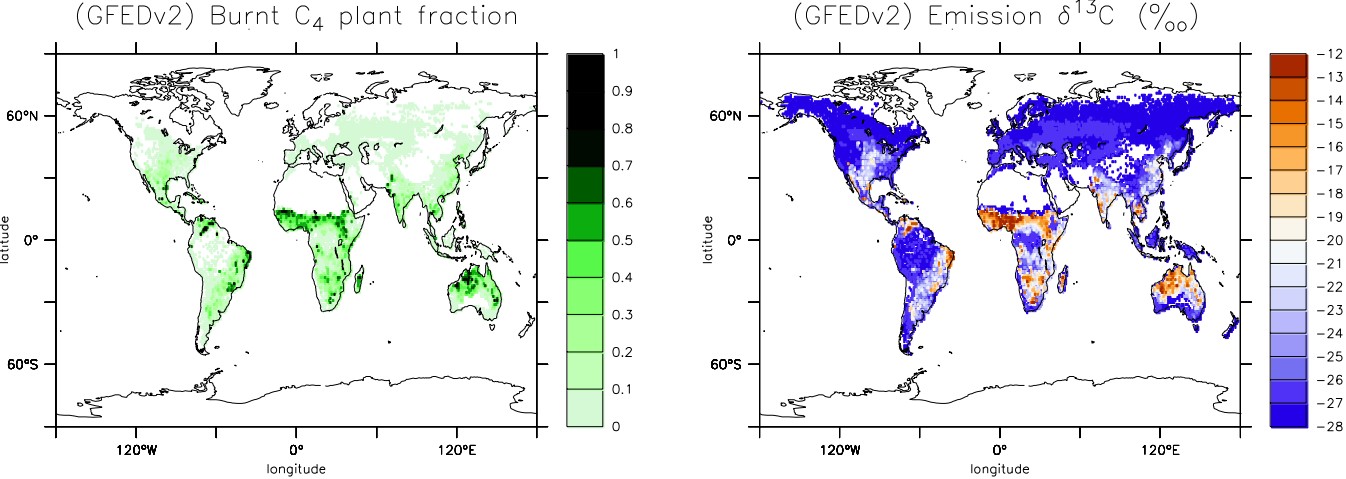

Fig. 3 Burnt biomass $C_4$ plant fraction (left) and corresponding isotopic signature of the emitted carbon (right) from GFED v2.1 database. Fields are "climatological" yearly averages (see text, also Fig. 4).

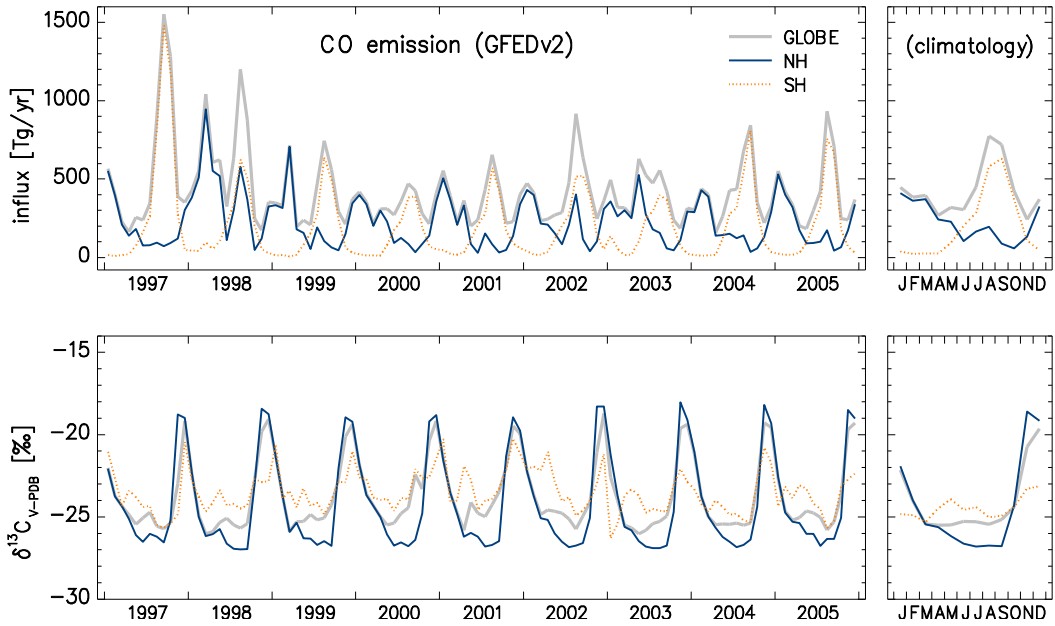

Fig. 4 Emission of CO from biomass burning sources based on the GFED v2.1 data. Upper: CO integrated flux in the northern (NH), southern hemispheres (SH) and globally. Lower: The carbon isotope composition of the respective fluxes. The right panels depict the "climatological" ensemble averages (shown in Fig. 3).





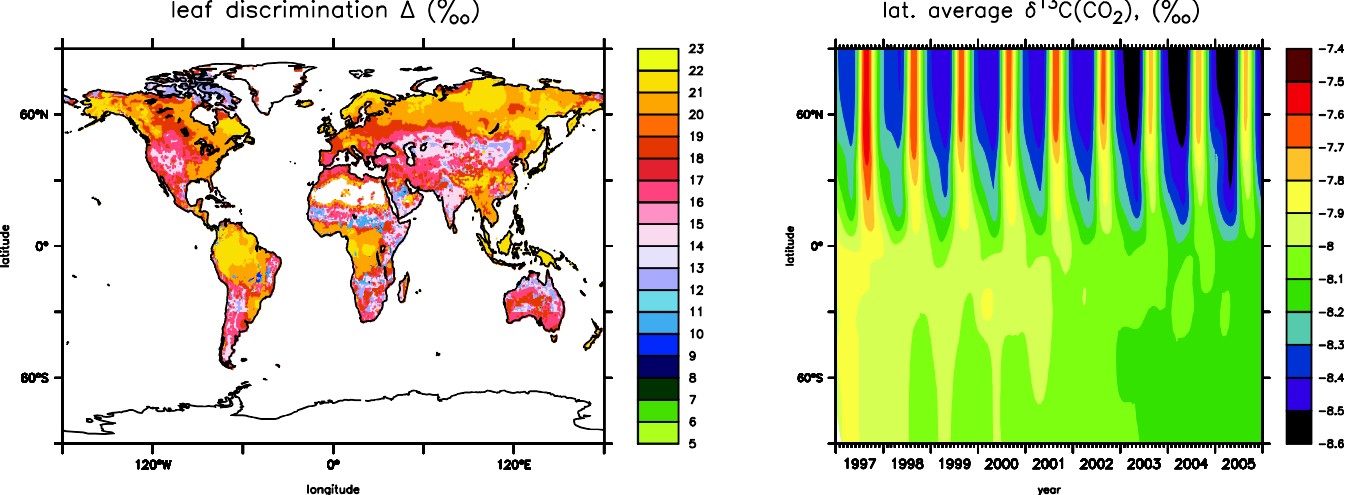

Fig. 5 Left: Global mean leaf discrimination distribution (ISOLUCP experiment, Scholze *et al.*, 2008). The distribution generally reflects the proportion of the $C_3/C_4$ metabolism and characteristic carbon photorespiratory fractionation in the various ecosystems, land use regimes and climate zones. Right: Time series of the latitudinal average surface isotopic composition of $CO_2$ from the GLOBALVIEW-CO2C13 (2009) data.







Fig. 7 Left: Annual CO emission from the surface sources (upper panel) and corresponding carbon isotopic composition (lower panel).
Right: Respective time series of zonal averages for the year 2000 emission with identical colour scale.



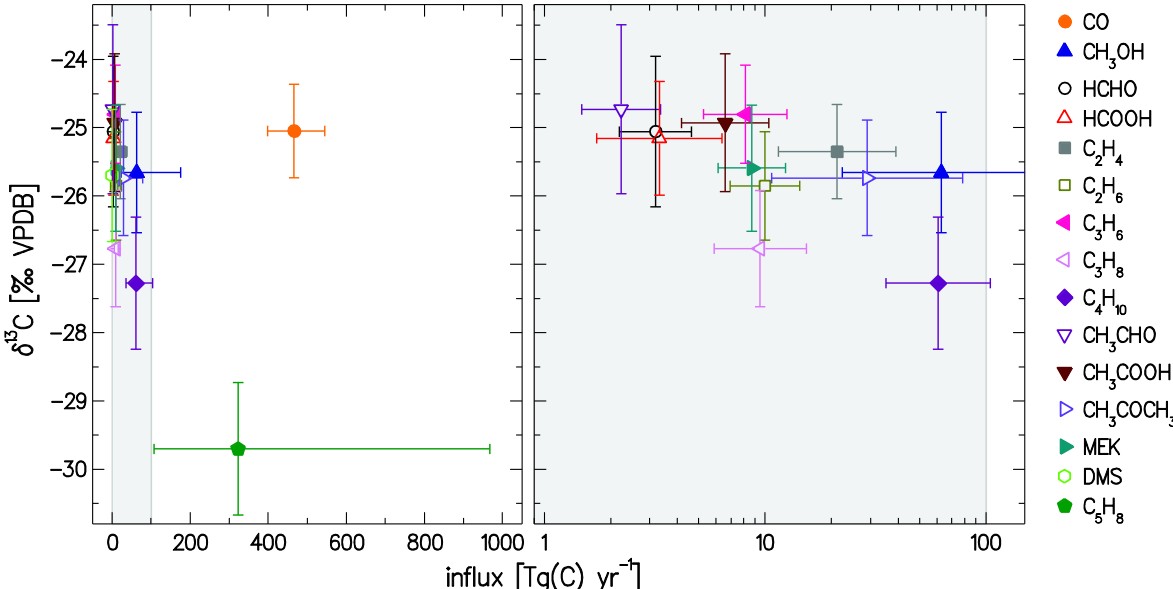

Fig. 8 Left: Overall annual surface emission isotopic compositions of the carbonaceous compounds. Right: Expanded shaded area in the left panel for the NMHCs/VOCs. The error bars refer to the uncertainty factors from Table 5 and are discussed in Sect. 2.2.



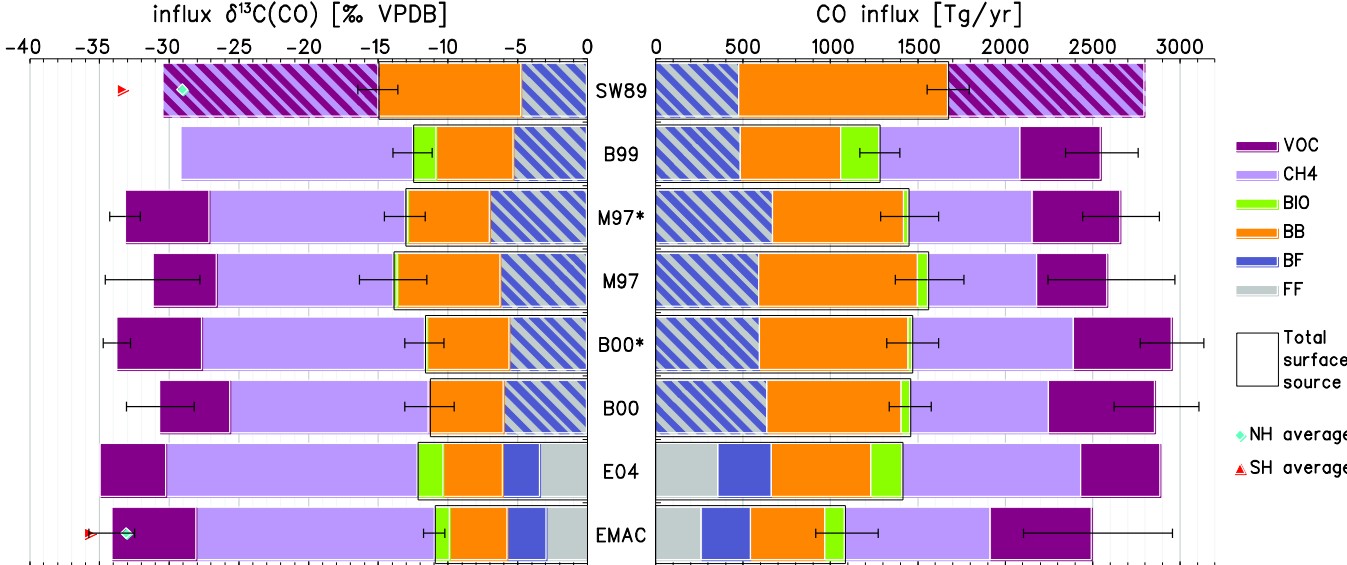

Fig. 9 Estimates of the tropospheric CO sources and their contribution to the overall source isotope composition from previous and the present studies (refers to Table 7). Abbreviations refer to: SW89 – Stevens and Wagner (1989); M97 – Manning *et al.* (1997) (case 2); B99 – Brenninkmeijer *et al.* (1999); B00 – Bergamaschi *et al.* (2000) (scenario S2); E04 – Emmons *et al.* (2004); EMAC – this study, setup based on EVAL$_2$, year 2000 (see text, Sect. 2). Asterisks denote *a priori* estimates of the corresponding inverse modelling studies. Note: Blue-grey hatched bars denote the aggregate of industrial emissions (FF and BF sources are not distinguished); SW89 report the total of photochemical sources only (light blue-violet hatched bars, respectively). Black frames denote the values for the total surface component. Right panel: Source terms by category. Left panel: Individual contribution of each source category to the overall source $\delta^{13}C(CO)$, calculated as a product of the share in total emission and respective source $\delta^{13}C$ average. Symbols denote the hemispheric tropospheric averages, where available.





**Tables**

Table 1  Description of EDGAR FT32 emission source sectors and associated isotopic signatures

| Category | Source sectors | Emission activity | δ¹³C [‰] |
|---|---|---|---|
| Biofuel combustion | B10, B20, B30, B40, B51 | industry, power generation, charcoal production, RCO[*], road transport | −25.0 [g] |
| Fuel combustion, production and transmission | F10, F20, F30, F40, F51, F54, F58, F60 [d], F80, F90 [c] | industry, power generation, conversion, RCO[*], road/non-road transport, international shipping, gas production | −27.5 |
| | F57 [a] | air traffic | −27.5 [f] |
| Industrial | I10, I20 [c] | iron and steel, non-ferrous metals | −27.5 [f] |
| | I30 [c], I60 [c], I70 [c], I90 [c], I50 [b] | chemicals, food/beverages/tobacco, solvents, misc. industry, pulp and paper | −27.5 [f] |
| Land use [b] | L41, L42, L44, L47 | (in)direct deforestation, savannah burning, vegetation fires | − [a] |
| | L43 | agricultural waste burning | −22.2 [e,g] |
| Waste [b] | W40, W50 [c] | waste incineration, misc. waste handling | −24.0 [f,g] |

Notes:
[a)] Excluded from the setup (or treated separately).
[b)] Assuming a biomass burning-related emission source.
[c)] Only CO emission (no VOCs).
[d)] Only VOCs emission (no CO).
[e)] For CO, a different signature of −21.3‰ is used (see text).
[f)] Fossil source assumed.
[g)] Reflects the relative contribution of $C_3$ and $C_4$ plant material.
[*] (Residential, Commercial and Other)

Table 2  Anthropogenic emission sources strengths and their isotopic signatures

| Species | Source [Tg(gas) yr⁻¹] | | | Totals | |
|---|---|---|---|---|---|
| | Biofuel | Fossils | Waste [a] | Emission [b] | δ¹³C [‰] |
| CO | 250.4 | 280.4 | 16.35 | 547.2 / 234.6 | −26.2 |
| $CH_3OH$ | 6.58 | 3.13 | 0.43 | 10.14 / 3.80 | −25.7 |
| HCHO | 3.50 | 0.98 | 0.23 | 4.71 / 1.88 | −25.5 |
| HCOOH | 3.56 | − | 0.23 | 3.79 / 0.99 | −24.9 |
| $C_2H_4$ | 5.11 | 3.54 | 0.34 | 8.99 / 7.70 | −26.0 |
| $C_2H_6$ | 2.91 | 6.11 | 0.19 | 9.21 / 7.36 | −26.6 |
| $C_3H_6$ | 2.28 | 1.49 | 0.15 | 3.92 / 3.36 | −26.1 |
| $C_3H_8$ | 0.91 | 9.45 | 0.06 | 10.42 / 8.51 | −27.2 |
| $C_4H_{10}$ | 1.16 | 70.67 | 0.08 | 71.91 / 59.44 | −27.4 |
| $CH_3CHO$ | 2.04 | − | 0.13 | 2.17 / 1.18 | −24.9 |
| $CH_3COOH$ | 6.52 | − | 0.43 | 6.95 / 2.78 | −24.9 |
| $CH_3COCH_3$ | 1.89 | 3.18 | 0.12 | 5.19 / 3.16 | −26.4 |
| MEK | 4.42 | 4.22 | 0.29 | 8.93 / 5.95 | −26.1 |

Notes:
[a)] Refers to the EDGAR sector L43.
[b)] Values are in [Tg(gas) yr⁻¹] / [Tg(C) yr⁻¹] units, respectively.



Table 3  Biomass burning emission sources strengths and their isotopic signatures

| Species | Source [Tg(gas) yr⁻¹] | | | Average $\delta^{13}C$ [‰] | |
|---|---|---|---|---|---|
| | NH | SH | Total [a] | NH | SH |
| CO [b] | 223.2 (170.8−396.7) | 202.8 (137.4−364.1) | 425.9 (336.8−589.9) / 182.6 (144.4−252.9) | −24.0 −(23.3−25.2) | −24.4 −(23.3−25.3) |
| CH₃OH | 3.17 | 2.98 | 6.15 / 2.31 | | |
| HCHO | 1.69 | 1.58 | 3.27 / 1.31 | | |
| HCOOH | 1.73 | 1.62 | 3.35 / 0.87 | | |
| C₂H₄ | 2.47 | 2.32 | 4.79 / 4.10 | | |
| C₂H₆ | 1.41 | 1.32 | 2.73 / 2.18 | | |
| C₃H₆ | 1.11 | 1.04 | 2.15 / 1.84 | −24.3 | −24.7 |
| C₃H₈ | 0.44 | 0.41 | 0.85 / 0.69 | | |
| C₄H₁₀ | 0.56 | 0.52 | 1.08 / 0.89 | | |
| CH₃CHO | 0.99 | 0.93 | 1.92 / 1.05 | | |
| CH₃COOH | 3.16 | 2.97 | 6.13 / 2.45 | | |
| CH₃COCH₃ | 0.91 | 0.86 | 1.77 / 1.08 | | |
| MEK | 2.14 | 2.00 | 4.14 / 2.76 | | |

Notes:
[a] Values are in [Tg(gas) yr⁻¹] and [Tg(C) yr⁻¹] units, respectively.
[b] For CO, interannual variation for 1997−2005 (monthly averages) is given in parentheses.

Table 4  Biogenic emission sources strengths and their isotopic signatures

| Species | Sources [Tg yr⁻¹] | | Totals | |
|---|---|---|---|---|
| | Land (Soils) | Ocean | Emission [a] | $\delta^{13}C$ [‰] |
| CO | 100.0 | 12.7 | 112.7 / 48.3 | −24.2 |
| CH₃OH | 151.0 [b] | − | 151.0 / 56.6 | −25.8 |
| HCOOH | 5.58 (1.65) | − | 5.58 / 1.46 | −25.4[c] |
| C₂H₄ | 10.0 (3.0) | 0.91 | 12.13 / 5.19 | −23.4 |
| C₂H₆ | − | 0.54 | 0.54 / 0.22 | −20.5 |
| C₃H₆ | 2.15 | 1.27 | 3.41 / 2.92 | −23.8 |
| C₃H₈ | − | 0.35 | 0.35 / 0.29 | −20.5 |
| C₄H₁₀ | − | 0.40 | 0.40 / 0.33 | −20.5 |
| CH₃COOH | 3.39 (1.44) | − | 3.39 / 1.36 | −25.7 |
| CH₃COCH₃ | 40.57 | − | 40.57 / 24.74 | −25.7 |
| DMS | 0.91 | − | 0.91 / 0.35 | −25.7 |
| Isoprene [d] | 346.03−385.35 | − | 346.03−385.35 / 305.07−339.73 | −28.6 to −27.2 |

Notes:
[a] Values are in [Tg(gas) yr⁻¹] and [Tg(C) yr⁻¹] units, respectively.
[b] Recommended updated value (Pozzer *et al.*, 2007).
[c] Corrected for emission from formicine ants (0.22 Tg yr⁻¹) of −19‰ (Johnson and Dawson, 1993).
[d] Calculated online.



Table 5  Surface emission sources in the EMAC (EVAL$_2$ setup)

| Species | Source [Tg(gas) yr$^{-1}$] | | | Aggregate uncertainty factor | Totals (uncertainty) | |
|---|---|---|---|---|---|---|
| | Anthropogenic (incl. Biofuel) | Biomass burning | Biogenic | | Emission [Tg(C) yr$^{-1}$] [a] | $\delta^{13}C$ [‰] |
| CO | 547.2 (250.4) | 425.9 | 112.7 | 1.17 | 465.6±79.1 | −25.0±0.7 |
| CH$_3$OH | 10.1 (6.6) | 6.15 | 151.0 | 2.81 | 62.7±113.2 | −25.7±0.9 |
| HCHO | 4.71 (3.50) | 3.27 | – | 1.45 | 3.2±1.5 | −25.1±1.1 |
| HCOOH | 3.79 (3.56) | 3.35 | 5.58 | 1.92 | 3.3±3.1 | −25.2±0.8 |
| C$_2$H$_4$ | 8.99 (5.11) | 4.79 | 10.9 | 1.84 | 21.1±17.9 | −25.3±0.7 |
| C$_2$H$_6$ | 9.21 (2.91) | 2.73 | 0.54 | 1.44 | 10.0±4.4 | −25.9±0.8 |
| C$_3$H$_6$ | 3.92 (2.28) | 2.15 | 3.42 | 1.54 | 8.1±4.4 | −24.8±0.7 |
| C$_3$H$_8$ | 10.4 (0.9) | 0.85 | 0.35 | 1.62 | 9.5±5.8 | −26.8±0.9 |
| C$_4$H$_{10}$ | 71.9 (1.2) | 1.08 | 0.40 | 1.72 | 60.7±43.8 | −27.3±1.0 |
| CH$_3$CHO | 2.17 (2.04) | 1.92 | – | 1.51 | 2.2±1.1 | −24.7±1.2 |
| CH$_3$COOH | 6.95 (6.52) | 6.13 | 3.39 | 1.58 | 6.6±3.8 | −24.9±1.0 |
| CH$_3$COCH$_3$ | 5.19 (1.89) | 1.77 | 40.6 | 2.71 | 29.0±49.6 | −25.7±0.8 |
| MEK | 8.93 (4.42) | 4.14 | – | 1.42 | 8.7±3.7 | −25.6±0.9 |
| DMS | – | – | 1.82 | 3.0 | 0.4±0.7 | −25.7±1.0 |
| C$_5$H$_8$ | – | – | 365.7 | 3.0 | 322.4±644.8 | −27.9±1.0 |

Notes:
[a]  Mind the different units used for individual categories and total values, *i.e.* [Tg(gas) yr$^{-1}$] and [Tg(C) yr$^{-1}$], respectively.

Table 6  Uncertainties associated with emission sources and isotopic signatures

| Category | Source | Emission ($\delta^{13}C$ signature) uncertainty [a] | | |
|---|---|---|---|---|
| | | CO | NMHCs/VOCs | Other [b] |
| Anthropogenic | Biofuel [c] | 2 (4.6‰) | 2 (4‰) | – |
| | Fossil fuel | 1.5 (0.3‰) [d] | 1.5−2.0 (2‰) [e] | – |
| | Waste [c] | 2 (4‰) | 2 (4‰) | – |
| Biogenic | Land (plants [f]) | 3 (1.9‰) | 3 (1.9‰) | 3 (1.9‰) |
| | Ocean | 2 (3.6‰) [g] | 2 (2‰) [h] | – |
| Biomass burning | | 1.3 (2‰) | 1.3 (2‰) | – |
| Pseudo-emission [i] | CH$_4$ | – | – | 0.04 % (0.05‰) [j] |
| | CO$_2$ | – | – | 0.03 % (0.02‰) |
| | CH$_3$Cl | – | – | 0.15 % (0.3‰) [k] |

Notes:
[a]  Given is the emission uncertainty factor (see Sect. 3.6) and isotopic signature uncertainty $\langle \delta_e \rangle$ (in parentheses).
[b]  Values assumed for biogenic isoprene, terrestrial DMS (plant emitted), and respective pseudo-emitted species.
[c]  C$_3$/C$_4$ plant composite, based on $\langle \delta^{13}C(C3) \rangle = 5.7$‰ and $\langle \delta^{13}C(C4) \rangle = 2.5$‰ (see text).
[d]  From Stevens *et al.* (1972).
[e]  Varies for each species due to the proportion of the fossil fuel (1.5) and industry (2.0) uncertainty factors contribution (Olivier *et al.*, 1999).
[f]  Derived from $\langle \delta^{13}C(CO_2) \rangle = 0.02$‰ and leaf discrimination uncertainty of $\langle \Delta \rangle = 2$‰.
[g]  Following Manning *et al.* (1997).
[h]  Based on variability in $\delta^{13}C$ of the marine carbon content from Avery Jr *et al.* (2006).
[i]  Quoted are mixing ratio uncertainties (not uncertainty factors).
[j]  Assigned equal to the upper limit of the atmospheric variation.
[k]  Error of the mean from Thompson *et al.* (2002).





Table 7  Tropospheric CO sources and their isotopic composition from the present and previous studies

| Study | SW89 | B99 | M97 [a,b] | B00 [a,b] | E04 [c] | EVAL2 [c] |
|---|---|---|---|---|---|---|
| Model | − | − | GFDL (2D) | TM2 | MOZART2 | **EMAC** |
| Emission inventories [d] | 1971 | 1972−1998 | 1987−1995 | 1987[+] | 1997−1999 | 2000[+] |
| $CH_4$ oxidation | (−55‰) [e] | 400−1000 (−52.6‰) | 624 (−52.6‰) | 795 (−51.1‰) | 1022 (−51‰) | 834 (−51.2‰) |
| NMHC oxidation | (−32.3‰) [e] | 200−600 (−32.2‰) | 403 (−29.3‰) | 607 (−23.9‰) | 453 (−30‰) | 579 (−26.1‰) |
| Fossil fuel / | 480 (−27.5‰) | 300−550 (−27.5‰) | 595 (−27‰) | 641 (−26.7‰) | 361 (−27‰) / | 272 (−27.4‰) / |
| Biofuel usage | | | | | 306 (−25‰) | 285 (−25‰) |
| Biomass burning | 1195 (−24‰) | 300−700 (−24.5‰) | 909 (−21‰) | 768 (−20‰) | 570 (−21.8‰) | 434 (−24.1‰) |
| Biogenic / | | 60−160 (−) / | −/ | −/ | 158 (−32‰) / | 102 (−25.7‰) / |
| Oceans | | 20−200 (−13.5‰) | 57 (−13.5‰) | 49 (5.1‰) | 20 (−12‰) | 13 (−13.5‰) |
| Photochemical sources | 1100−1250 (−38.4‰) [e] | 1265 (−33.5‰) [f] | 1027 (−43.4‰) | 1402 (−39.3‰) | 1475 (−44.6‰) | 1414 (−40.9‰) |
| Uncertainty | ±125 (±1.7‰) | ±180 (±3.7‰) | ±182 (±3.5‰) | ±127 (±2.5‰) | − | ±420 (±4.4‰) |
| **Surface sources** | **1550−1700 (−25.0‰)** | **1285 (−24.8‰)** [f] | **1561 (−23‰)** | **1458 (−22.1‰)** | **1415 (−24.8‰)** | **1086 (−25.2‰)** |
| **Uncertainty** | **±125 (±1.7‰)** | **±238 (±1.4‰)** | **±207 (±2.4‰)** | **±125 (±1.8‰)** | **−** | **±194 (±0.7‰)** |
| Total sources | 2800 (−30.3‰) | 2550 (−34.9‰) | 2588 (−31.1‰) | 2860 (−30.5‰) | 2890 (−34.9‰) | 2525 (−34.1‰) |
| Overall uncertainty | ±250 (±2.0‰) | ±216 (±1.4‰) | ±389 (±3.4‰) | ±252 (±2.4‰) | − | ±462 (±1.6‰) |

Notes: The source/sink terms are given in [Tg(CO) yr$^{-1}$] with the corresponding $\delta^{13}C$ composition [‰ V-PDB] of the sources in parentheses. Values are the tropospheric averages. Abbreviations refer to: SW89 – Stevens and Wagner (1989); M97 – Manning *et al.* (1997) (case 2); B99 – Brenninkmeijer *et al.* (1999); B00 – Bergamaschi *et al.* (2000) (scenario S2); E04 – Emmons *et al.* (2004); EVAL2 – this study (see Sects. 1, 2).

[a] A simplified chemistry scheme (no intermediates in the $CH_4 \rightarrow CO$ chain, no NMHC chemistry) is used.

[b] An inversion technique to improve the emission strengths/isotope signatures is employed.

[c] A detailed chemistry scheme (*e.g.*, $CH_4$ and NMHC chemistry with intermediates and removal processes) is used.

[d] The year(s) the aggregate of the emission inventories correspond closest to; the plus signs indicate that the transient biomass burning inventory was used, with the listed year referring to the anthropogenic emissions revision.

[e] The authors assume a too high NMHC:$CH_4$ source fluxes partitioning of 5.5 based on then limited information on sources O isotope composition. The $^{13}C$ mass-balance and photochemical source is reanalysed here in light of current knowledge on the $\delta^{18}O$ signatures of CO sources (see, *e.g.*, B99).

[f] The average signature results from the respective source terms (denoted as the sum) assumed within the given limits.