# Peer review of "Uncertainties of fluxes and 13C/12C ratios of atmospheric reactive gases emissions"

_Atmospheric Chemistry and Physics, 2016_

## Author Comment (AC1) · 10 Jan 2017

**Corrigendum**

Dear Editor, dear Reader,

Unfortunately, the software used to produce this manuscript is suffering from typesetting errors for formulae.

Thus, in the published manuscript, some formulae parentheses are missing, rendering Eqs. (1), (11), (12) and (13) erroneous. Please, refer to this corrigendum for the correct formulations. These are:

$$^{\text{rare},i}f = \frac{^{\text{rare},i}R \cdot q}{1 + (1-q) \cdot \sum_j {^{\text{rare},j}R}}, \quad {^i R} = \left(\delta^i + 1\right) \cdot {^i R_{\text{st}}} . \tag{1}$$

$$\langle {^i R_e} \rangle = \sum_s \left( \left| \frac{\partial {^i R_e}}{\partial F_s} \right| \cdot \langle F_s \rangle + \left| \frac{\partial {^i R_e}}{\partial R_s} \right| \cdot \langle {^i R_s} \rangle \right) =$$
$$= \sum_s \left( \left| \varphi^2 \cdot \sum_n F_n \left( {^i R_s} - {^i R_n} \right) \right| \cdot \langle F_s \rangle + \left| \varphi \cdot F_s \right| \cdot \langle {^i R_s} \rangle \right) \tag{11}$$

$$\langle {^i R_e} \rangle = \sqrt{ \sum_s \left( \left[ \varphi^2 \cdot \sum_n F_n \left( {^i R_s} - {^i R_n} \right) \right]^2 \cdot \langle F_s \rangle^2 + \left( \varphi \cdot F_s \right)^2 \cdot \langle {^i R_s} \rangle^2 \right) } . \tag{12}$$

$$\begin{cases} \dfrac{^{13\text{C}}F}{F} = \left(1 - f_{\text{C}_4}\right) \dfrac{q \cdot R_{\text{C}_3}}{R_{\text{C}_3} + 1} + f_{\text{C}_4} \dfrac{q \cdot R_{\text{C}_4}}{R_{\text{C}_4} + 1} \\[3mm] \dfrac{^{12\text{C}}F}{F} = \left(1 - f_{\text{C}_4}\right) \dfrac{(1-q) \cdot R_{\text{C}_3} + 1}{R_{\text{C}_3} + 1} + \\[3mm] \qquad + f_{\text{C}_4} \dfrac{(1-q) \cdot R_{\text{C}_4} + 1}{R_{\text{C}_4} + 1} \end{cases} \tag{13}$$

With apologies, on behalf of all authors,

S. Gromov

---

## Author Comment (AC2) · 26 Jan 2017

Dear Editor, dear Reader,

Unfortunately, another typo was identified in Eq. (1) of the manuscript. An extraneous term (q−1) should be removed from the denominator. The correct formulation should read:

$$^{\mathsf{rare},i}f = \frac{^{\mathsf{rare},i}R \cdot q}{1 + \sum_j{}^{\mathsf{rare},j}R}, \quad ^iR = \left(\delta^i + 1\right) \cdot {}^iR_{\mathsf{st}} \qquad (1)$$

We are indebted to Franziska Frank (IPA, DLR Oberpfaffenhofen) for pointing out this error.

[Figure]

With apologies, on behalf of all authors,

S. Gromov

---

## Referee Comment (RC1) · Anonymous Referee #1 · 17 Feb 2017

Review on:
**Proxies and uncertainties for 13C/12C ratios of atmospheric reactive
gases emissions-** by Gromov et al.

The paper presents a review of the proxy data on stable carbon isotope ratios and uncertainties of emissions of reactive carbonaceous compounds into the atmosphere, with a focus on CO sources, with the goal to be used in the global modelling of isotope ratio distribution. This is a further valuable contribution to the hitherto scarce studies on in this field. Isotopes deliver important adjunct information which can increase the understanding of the pollution sources and atmospheric processes. Therefore the paper is highly suitable to be published in the journal.
The paper contains yet some weak points which need to be improved before publishing.

Specific comments
Generally, a discussion on the benefits of using isotopes in the atmospheric research is missing. This would be beneficial to convince the reader regarding the impact of this paper. Moreover, it would ease the final discussion on the few trials in the past to use a 3D chemical model to interpret the global distribution of the ethane isotopic composition (I will come back on that). I also see this paper as a perfect platform to discuss the possibilities and current limitations of using isotope ratio measurements for the purpose of gaining additional insight into the source apportionment. The reader shouldn't get the impression, that it is absolutely not important which source delta values are used in the model input, since in the end, due to the emission fluxes, all diversity is anyway flattened out.

**Section2.1:**
- The paragraph on Page4Lines101to 113 should be revised: if so detailed, then it should be done up to the end (i.e. information on the old and 'new' scale, the PDP and V-PDP 13C/13C isotope ratios, cite IUPAC paper Brandt et al. 2010). For consistency, the authors should consider to use the same notation in the sentence on Lines 107 to 109 (either 'per mil'or '‰').
- Equation 1: define j. Moreover, - this is a problem of taste – is it necessary to sum the isotopologues multiply carrying a 13C for this relatively low molecular compounds? The error induced when you don't account the **natural** isoprene with five heavy isotopes is definitively insignificant compared to all other sources of uncertainties.

**Section3:**
This Section should be thoroughly revised. Generally, the readability is not optimal. There are multiple points to be take care of:
- There are too many details which are nowhere else used, such as information on CAM plants or ethyne. Shorten and make it more concise!
- In the same direction: comparing source delta values is at some places very confusing. The reader doesn't get always the information, is it CO, CO2, a single organic compound, total carbon? As example: Paragraph starting on Page8Line 212: Stevens et al. present total carbon, the rest of the literature is dedicated specifically to CO.
- Page8Line220: from Fig5 in Popa et al., I see a much lower CO delta zero (ca. -29‰)?
- There are cases where the emissions during biomass burning are similar to the parent fuel, not always (see resulting single compounds, Gensch et al. 2014). As a suggestion, this might be the place to make the understanding easier for the reader (e.g. by discussing the accompanying processes and their isotopic fractionation, which is more significant for the reactions of thermal decomposition (KIE) than the one occurring only by evaporation of the compounds of interest from the plant tissues).

- The paper gives a very useful tool to calculate source delta values, where the literature information is missing. On the other side, the authors don't make use of the existing source measurements. For VOC source delta, they should use (and cite) the newer review on using isotopes in the atmospheric research: Gensch et al. 2014.
- Paragraph starting on Page10Line278 should be moved into the Section 3.2.

**Section 4.2.**

should be completely revised. First of all, Stein et al. 2007 is not the only trial to globally model isotopes. Saito et al. 2011 should be definitively cited here! This will be beneficial for the authors to discuss the importance of a detailed study, using the whole diversity of information the isotope community is owing up to date. Both studies are very good examples of that. The former shows that there might be anthropogenic sources missing in the inventories. This conclusion cannot be gained from concentration measurements. The latter discuss the importance of having accurate KIE of the atmospheric degradation reactions.

Other comments:
Page1Lines25to27: reformulate 'The situation complicates, if the isotope-resolved emissions (i.e., fluxes separated using the information on the isotope ratios of the emitted compounds) are to be used. '

Editorial revisions:
Page2Line49: 'revises' instead of 'revisits'
Page3Line67: 'set' instead of 'determine'
Page13Line372: better 'e.g.' than 'viz.'
Generally: 'and' instead '/' (e.g. Page3Line73 'isoprene and monoterpenes' instead of 'isoprene/monoterpenes')
Replace 'escorting' by e.g. 'associated with' (two times).

---

## Short Comment (SC1) · 17 Feb 2017

Dear authors,

first of all let me congratulate for this excellent manuscript. I very much appreciate the huge efforts you have made for putting this work together. It certainly will provide us with a better understanding of the global CO budget when applying stable carbon isotopes. However, it also indicates the limitations and uncertainties when applying a stable isotope approach.

I have one major issue regarding the stable carbon isotope source signatures of methanol derived from vegetation which I hope the authors consider in their revised manuscript. In Table 4 'Biogenic emission sources strengths and their isotopic signatures' the stable isotope values has a value of -25.8 ‰ . The authors might be not

aware of some recent studies by Keppler et al. (2004) and Giebel et al. (2010) which clearly show that methanol emissions from living plants and from combustion are considerably more negative than provided by Gromov an co-authors. Relative to the bulk biomass of plants, the carbon isotope fractionation exhibited by the plant methoxyl pool - which is definitely the major carbon source of methanol emitted from living plants - is very large. Methoxyl groups in the plant kingdom are exceptionally depleted in 13C and thus plant-derived C1 volatile organic compounds such as methanol have drastically depleted stable carbon isotope values. The range provided by Keppler et al. 2004, was -50 to -85 ‰ . A similar range was measured by Giebel et al. (2010). Thus I would like to suggest that the authors update their manuscript with this data but also use them for their model simulations.

With regards to the application of delta notation (see also comment by reviewer 1 "either per mil or ‰ ") I have an alternative suggestion. To comply with guidelines for the International System of Units (SI), the authors might follow the recent proposal of Brand and Coplen (2012) and use the term urey, after H.C. Urey (symbol Ur), as the isotope delta value unit. In such a manner, an isotope-delta value expressed traditionally as $-25$ ‰ can be written $-25$ mUr. However, this might be a matter of taste.

References:

Keppler, F., Kalin, R.M., Harper, D.B., McRoberts, W.C. and Hamilton, J.T.G. (2004) Carbon isotope anomaly in the major plant C1 pool and its global biogeochemical implications. Biogeosciences 1, 123-131.

Giebel, B.M., Swart, P.K. and Riemer, D.D. (2010) $\delta$13C Stable Isotope Analysis of Atmospheric Oxygenated Volatile Organic Compounds by Gas Chromatography-Isotope Ratio Mass Spectrometry. Anal. Chem. 82, 6797-6806.

---

## Referee Comment (RC2) · Anonymous Referee #2 · 13 Mar 2017

This paper presents a comprehensive synthesis of the carbon isotopic composition sources of CO and some hydrocarbons to the atmosphere. It is very useful work and provides a reference dataset (similar to emissions databases) for future model studies with isotope-enabled models. Most of my remarks are related to the presentation. One issue is uncommon usage of constructions or terms, which made the manuscript hard to read for me (see many detailed points below). At some places, I could not follow the argumentation based on the information presented in the figures (some of the figures may be incomplete?). A few general comments related to the handling of errors. The authors are encouraged to help the reader follow their argumentation at some places, where the link between results and scientific interpretation is not straightforward.

General points:

1) Since the authors make a strong and valid point on the value of errors, I was surprised that their individual budget estimates in section 3 do not come with errors

2) Figures: I wonder whether Figures 6a and 7a are shown correctly. There are hardly any emissions in Africa. This does not look OK. There are rather large biogenic emissions from Indonesia in Fig 6a but the isotopic composition is the one of the oceanic source. Can that be true? Also see comment below on the origin of the oceanic value of 13 permil.

3) Tables: there are at least two errors in the conversion from Tg(gas) to TgC, for C2H4 and C2H6 in Table 4. Please check the other values carefully. Table 5: Several points are not clear to me: a) how is the aggregate uncertainty factor derived specifically? b) What is the relation between columns 2,3,4 (individual surface sources) and column 6. c) Column 6 seems way too low as total surface emission for CO. d) Can you comment on the huge error bar for the isoprene emissions? Table 6: Is the uncertainty for CH3Cl isotopic composition really that low?

4) Part of the discussion (L630 ff) is very hard to follow and in my view does not reflect the results shown in the figures, see detailed points below. Another part is L672 ff (last paragraph of section 4.1). The conclusions that the authors are hard to related to the results shown in the figures.

5) The issue that inverse models report too optimistic posteriori errors for the combined source is quite a strong statement. I wonder whether this is not a misunderstanding by the authors. According to my knowledge inverse modeling involves rather solid error calculation, and especially in inverse modeling the constraint on the total source (from mole fraction observations) is much more tight than the sum of the individual components. I suggest that the authors contact with an inverse modeler to check this.

Scientific and presentation issues:

Title: This is in my view an uncommon usage of the term proxies. Does it need to be

in the title?

Line 27 "which factors determine a particular emission source isotope ratios". The authors are encouraged to avoid such dense constructions with multiple nouns. For a reader, it is much easier to grasp constructions with "of". In this case "which factors determine the isotope ratio of a particular emission source". This could be simplified at many places throughout the paper: Here, there is also an "s" too much. Also, in the next sentence, please specify "the latter"

L47: see comment L27, too many nouns. . .

L58: "Tendencies" is a rather unspecific term. Why not concentrations?

L58/59: "by modifying its vertical diffusive flux boundary conditions at the lowest model layer" is very technical. Change to "by adding emissions to the . . ."

L86: For me "isotope separation" is a bit strange term in this context. Why not: Fluxes of individual isotopologues?

L 92: You could help the reader with an example calculation of Eq 1. E.g. for the case of CO with one 13C and one 18O atom. How do the ratios, deltas and fluxes behave?

L103-109: This is very hard to follow, please reformulate. Avoid "ex post facto".

L142 ff: please rewrite sentence that includes Eq 5.

L150 ff: If the rare isotopologue fluxes are off by 1% and the abundant isotopologue flux is correct, the isotope ratios would be wrong by 10 permil. Please comment/clarify

L166: I cannot understand the remark on the uncertainties of guessed parameters. When you guess a parameter, you can often also guess an uncertainty.

L275: what about the sensitivity in other regions of the world (Africa, South America, Asia)?

L306: where does the ratio 250/280 come from?

L353: Cryptic sentence about the use of a different proxy for CO in GFED. Not clear to me

L382 and Fig 6: It is not clear for me how you come from the -20.5 for the marine carbon content in the text to the -13 in Fig 6 for oceanic CO.

L 480/1 and Fig 6. It is not visible in Fig 6 that NH terrestrial sources are smaller than oceanic ones in winter. Fig 6 implies that oceanic sources are zero?

L490: What causes the range in the isoprene emission?

L501: No emissions in Africa in Fig 7 (see above).

L541: why is photolysis of chloromethane included as isotope resolved processes

L570/1: Either number or plant type is wrong in the example.

L582: Where do the UF estimates come from?

L592: Where does the biofuel uncertainty come from? A table, or is this additional information?

L605: do not let the reader guess which of the studies are bottom-up.

Line 630 - 658: This part is not clear. I have a hard time following the arguments and finding back in the figures what the authors describe.

a) B00 does not really seem to have a much lower Ch4 derived CO source than B99 or SM89 (line 630)

b) the a posteriori sources . . . (line 633). The logic is wrong or at least not clear. I do not see that a posteriori sources are reduced in M97.

c) (l634) I do not see that B00 decreases CH4 derived CO less than M97. In the posteriori results the BB source also increases. The description is not clear at all, and it is also not clear what the bottom line should be.

[Figure]

d) L635: These two studies do NTO show the largest BB emissions, this is the case for SW89.

e) L645: please help the reader why and how the cold start issue could be addressed by 18O but not by 13C

f) L646: It is not immediately clear that the strengths of other sources can be constrained better with isotopes.

Technical issues:

L29-30: ... how comprehensive should the model be

L46: It is not clear what "in the evaluation setup" means. Is this necessary?

L61: at THE respective

L67: specific emission category

L74: delete "THE"

L75: simplify: . . . , which leads to more realistic. . ..

L79: the OBSERVED mixing

L87 regular → total

L114: not sure what these preparation tools are. Leave out?

L115: avoid double plurals (fluxes values –? flux values), also in several other places.

L124: leave out "superposed"

L125: clearly comprehensible → clear

L126: or various isotope mixtures → with different isotopic composition

L127: abundance → source strength

L129: Leave out "To give an example". This is not an example.

L136: avoid "impermeable"

L136/7: "because in contrast to ratios, it is much more difficult to relate" → because it is difficult to relate

L164: rewrite/explain "by fitting their (isotope mass-balanced) sum to the given integral."

L168 with → using

L176: add ISOTOPE ratio

L186/7. Leave out this sentence, it creates more confusion that clarification. (Why would it?)

L188 the → a

L202: "surface and adjacent layers" is unspecific. I think you mean the first and second model layer, correct?

L203: specify the remaining sectors

L206: and OTHER emitted

L262: significantLY HIGHER

L272: define "bio-petrol"

L315: use different word for "superincumbent" → higher?

L320 verb missing (is)

L329: second MODEL layer

L338: El Nino Southern Oscillation (ENSO) climate pattern

L341/2: The variation of the d13C of the emission flux is. . .

L401: replace "tolerating"

L436: replace "escorted"

L437: replace "rational"

L445/6: Rewrite: "may be used as a proxy for the average bulk leaf biomass, thus concluding the depletion of the emitted isoprene in relation to it."

L459: framework DEVELOPED by

L461: a set of numerous parameters

L477: replace perceptibly by more quantitative term, or leave out

L478: . . . which results in

L495: Avoid double plural (trace gas emissions)

L501 & 503: the largest . . . . A comparable. Please modify.

L530: . . .sources associated with biogenic activity that emit isotopically light methane

L531: corresponding TO

L540: replace "isotopic carbon"

L562: In contrast, uncertainties of isotope signatures are reported . . .

L565/6; rewrite sentence

L583: avoid double plural

L620: One infers a similar. . .

L650: Replace/leave out "inquiries"

---

## Author Response (AR1)

S. Gromov, on behalf of all authors (sergey.gromov@mpic.de)

Authors apologise for the delayed reply, which was partly caused by some recalculations we had to perform after receiving valuable and constructive comments. We appreciate a very honest assessment by the Referees and notice a lot of work which Referee 2 put in his review. We are happy to incorporate all the improvements suggested (with an exception of few issues on presentation style). We furthermore apologise for the technical issues with formulation and figures in the initial manuscript which we have eliminated in the revised version. Note that a new title of the manuscript is proposed. Below we provide the answers (coloured green) to all comments received, followed by the marked-up version of the revised manuscript.

**Referee 1**

The paper presents a review of the proxy data on stable carbon isotope ratios and uncertainties of emissions of reactive carbonaceous compounds into the atmosphere, with a focus on CO sources, with the goal to be used in the global modelling of isotope ratio distribution. This is a further valuable contribution to the hitherto scarce studies on in this field. Isotopes deliver important adjunct information which can increase the understanding of the pollution sources and atmospheric processes. Therefore the paper is highly suitable to be published in the journal.

The paper contains yet some weak points which need to be improved before publishing.

**Specific comments**

Generally, a discussion on the benefits of using isotopes in the atmospheric research is missing. This would be beneficial to convince the reader regarding the impact of this paper. Moreover, it would ease the final discussion on the few trials in the past to use a 3D chemical model to interpret the global distribution of the ethane isotopic composition (I will come back on that).

This certainly is an issue and we therefore now will refer to Brenninkmeijer, Janssen, Kaiser, Röckmann, Goldstein and Shaw, Gensch, who at least review the benefits of using stable isotope ratios for atmospheric trace gases we consider in this study. We think that the current paper is more a technical paper and the proof of the pudding will be in the eating. In other words, we cannot use the current paper to convince the reader regarding its impact. For colleagues to engage in measuring stable isotope ratios on atmospheric trace gases, the merits may differ from case to case, and a blanket blessing we cannot deliver, whereas a more detailed assessment is beyond the scope of this paper. Concerning CO itself we referred to the paper by Brenninkmeijer, Röckmann, Bräunlich and others.

I also see this paper as a perfect platform to discuss the possibilities and current limitations of using isotope ratio measurements for the purpose of gaining additional insight into the source apportionment. The reader shouldn't get the impression, that it is absolutely not important which source delta values are used in the model input, since in the end, due to the emission fluxes, all diversity is anyway flattened out.

These are two issues. Although there is indeed a need for such a discussion on "sostisomitracheme" (sources, stable, isotopes, mixing, transport, chemistry and measurement) such would constitute a paper on its own. Second, for not falsely generating the impression, alerted to by the reviewer, that source delta values used in the model input are absolutely not important, since in the end - due to the emission fluxes - all diversity is anyway flattened out, we have improved sections of the manuscript that may falsely generate that impression.

**Section2.1:**

- The paragraph on Page4Lines101to 113 should be revised: if so detailed, then it should be done up to the end (i.e. information on the old and 'new' scale, the PDP and V-PDP 13C/13C isotope ratios, cite IUPAC paper Brandt et al. 2010).

We see a point of moving this detail to the end, although it is only one paragraph. The reason why we discuss this issue right here, while presenting the mathematical formulation, (and would like to keep it this way) is that we want to avoid confusion about the scale issues that are often central to recent discussions (for instance some colleagues refer to permil as a unit, although like %, ‰ it is not a unit). In the model we deal with real ratios and to avoid ambiguities, we inform the reader about which $^{13}C/^{12}C$ ratio we have used. As the referee suggests, we now include a reference to IUPAC report by Brand and Coplen (Brand *et al.*, 2010).

For consistency, the authors should consider to use the same notation in the sentence on Lines 107 to 109 (either 'per mil' or '‰').

We have removed these inconsistencies.

- Equation 1: define j.

Apologies, this got lost during editing. We add: "…, index $j$ cycles all rare isotopologues (*e.g.*, $^{13}CO$ for stable C, $C^{17}O$ and $C^{18}O$ for stable O substitutions of CO), …"

Moreover, - this is a problem of taste – is it necessary to sum the isotopologues multiply carrying a 13C for this relatively low molecular compounds? The error induced when you don't account the **natural** isoprene with five heavy isotopes is definitively insignificant compared to all other sources of uncertainties.

This is correct; multiple substitutions are insignificant in this sense. Indeed, we do not account for these by considering single substitutions only. Because the index $j$ was not explained (see the comment above), the sense of summing in Eq. 1 was not clear. Essentially we account for all abundant (i.e. $^{12}C$ or $^{16}O$) isotopes that are present in rare isotopologues bearing more than one element of interest. These are, for instance, the four $^{12}C$ atoms in a singly $^{13}C$-substituted isotopologue of $C_5H_8$. This is also implied by "multiple rare isotopes" mentioned at Line 95.

**Section3:**

This Section should be thoroughly revised.

Thank you for this advice. We have now modified this section w.r.t. content and formulations, also with a good deal of improvements suggested by Referee #2.

Generally, the readability is not optimal. There are multiple points to be take care of:

- There are too many details which are nowhere else used, such as information on CAM plants or ethyne. Shorten and make it more concise!

The purpose of including information that is not relevant for the given emission inventory is to acquaint the Reader with the widest range of options he/she may encounter whilst dealing with a newer emission inventory. The latter, in contrast to EDGAR or OLSEN, may contain information on ethyne emission fluxes or CAM plants distribution. We there fore would like to keep these details.

- In the same direction: comparing source delta values is at some places very confusing. The reader doesn't get always the information, is it CO, CO2, a single organic compound, total carbon? As

example: Paragraph starting on Page 8 Line 212: Stevens et al. present total carbon, the rest of the literature is dedicated specifically to CO.

We took care to report only the isotope ratios measured in species being discussed (we add a statement otherwise), and check for that in the revised manuscript. Furthermore, Stevens *et al.* (1972) explicitly report measurements for CO (we could not find 'total carbon' mentioned). However, since we discuss the fossil fuel signature for NMHCs/VOCs below, we remove "… and other NMHCs/VOCs" to avoid ambiguity here.

- Page8Line220: from Fig5 in Popa et al., I see a much lower CO delta zero (ca. -29‰)?

We made a deeper analysis of the data from Popa *et al.* (2014), as it presents two separate mixing cases, that is, at the entrance and at the exit of the tunnel, respectively (see the Figure below). For the "entrance" case, the background component is lighter in $^{13}$C, which is likely due to large share of CO produced from CH$_4$/VOCs oxidation (sampling was done in June; this CO is also lighter in $^{18}$O). In the "exit" case, the heavier in $^{13}$C "background" component is likely the CO produced from higher HCs of the same exhaust plume (plus fractionation), as well as the sink fractionation in self- or catalytic CO destruction, as authors explicate. In both cases, the "keeling plot" analysis points at similar signatures of the admixed (emitted by traffic) CO. The average of these two signatures (assumed being uncorrelated estimates) was quoted in the manuscript.

[Figure]

- There are cases where the emissions during biomass burning are similar to the parent fuel, not always (see resulting single compounds, Gensch *et al.*, 2014). As a suggestion, this might be the place to make the understanding easier for the reader (e.g. by discussing the accompanying processes and their isotopic fractionation, which is more significant for the reactions of thermal decomposition (KIE) than the one occurring only by evaporation of the compounds of interest from the plant tissues

Thank you, this work was overlooked on our side and is indeed a useful overview to which we refer in the revised manuscript.

**F. Keppler**

I have one major issue regarding the stable carbon isotope source signatures of methanol derived from vegetation which I hope the authors consider in their revised manuscript. In Table 4 'Biogenic emission sources strengths and their isotopic signatures' the stable isotope values has a value of -25.8 ‰. The authors might be not aware of some recent studies by Keppler et al. (2004) and Giebel et al. (2010) which clearly show that methanol emissions from living plants and from combustion are considerably more negative than provided by Gromov an co-authors. Relative to the bulk biomass of plants, the carbon isotope fractionation exhibited by the plant methoxyl pool - which is definitely the major carbon source of methanol emitted from living plants - is very large. Methoxyl groups in the plant kingdom are exceptionally depleted in 13C and thus plant-derived C1 volatile organic compounds such as methanol have drastically depleted stable carbon isotope values. The range provided by Keppler et al. 2004, was -50 to -85 ‰. A similar range was measured by Giebel et al. (2010). Thus I would like to suggest that the authors update their manuscript with this data but also use them for their model simulations.

We are indebted to Frank Keppler for pointing this out. At an early stage of compiling our emission inventory we have overlooked these important studies. We have recalculated the $\delta^{13}C$ of $CH_3OH$ emissions from plants and biomass burning and update the integrals in the revised manuscript. Importantly, the input of more $^{13}C$-depleted methanol aggravates the issue of missing global $^{13}CO$ emission that we emphasise.

With regards to the application of delta notation (see also comment by reviewer 1 "either per mil or ‰") I have an alternative suggestion. To comply with guidelines for the International System of Units (SI), the authors might follow the recent proposal of Brand and Coplen (2012) and use the term urey, after H.C. Urey (symbol Ur), as the isotope delta value unit. In such a manner, an isotope-delta value expressed traditionally as −25 ‰ can be written −25 mUr. However, this might be a matter of taste.

We prefer to keep the notation used.

**Referee 2**

This paper presents a comprehensive synthesis of the carbon isotopic composition sources of CO and some hydrocarbons to the atmosphere. It is very useful work and provides a reference dataset (similar to emissions databases) for future model studies with isotope-enabled models.

Most of my remarks are related to the presentation.

One issue is uncommon usage of constructions or terms, which made the manuscript hard to read for me (see many detailed points below). At some places, I could not follow the argumentation based on the information presented in the figures (some of the figures may be incomplete?). A few general comments related to the handling of errors. The authors are encouraged to help the reader follow their argumentation at some places, where the link between results and scientific interpretation is not straightforward.

**General points:**

1) Since the authors make a strong and valid point on the value of errors, I was surprised that their individual budget estimates in section 3 do not come with errors

We, in turn, are surprised by this comment. One of the main intentions of this manuscript is to deliver new, better uncertainty estimates. Tables 5 through 7 (to which we refer in Section 3 and further) contain error estimates for our emission setup. Table 6 is dedicated to uncertainties of flux and isotope ratios of individual emission categories (quoted in Tables 3–5 they would be presented redundantly). Perhaps, you imply that we did not include the "±" notation in the text of the manuscript; this is done in order to improve its readability and under assumption that the reader can refer to the tables at hand. You may also like to revisit paragraph [47] in Section 3, where our budget estimate is presented with uncertainties about the flux and $\delta^{13}C$ of CO.

2) Figures: I wonder whether Figures 6a and 7a are shown correctly. There are hardly any emissions in Africa. This does not look OK. There are rather large biogenic emissions from Indonesia in Fig 6a but the isotopic composition is the one of the oceanic source. Can that be true? Also see comment below on the origin of the oceanic value of 13 permil.

Thank you for pointing this out. Unfortunately, this is a rendering error of the software used to produce the PDF file containing the manuscript. Large emission over Africa is seen in the original figure (and covariates with the emission $\delta^{13}C$ seen nonetheless in the lower panels). We took care to free the final submitted manuscript from this production defect.

3) Tables: there are at least two errors in the conversion from Tg(gas) to TgC, for C2H4 and C2H6 in Table 4. Please check the other values carefully.

We do not see the mistake here, perhaps we are missing something. The conversion factor from Tg(gas) to Tg(C) should equal $\gamma$=24.02/28.05=0.8563 (here we use molar masses of $C_2H_4$ and C rounded up to 0.01). Taking the $C_2H_4$ BB emission of 4.79 Tg(gas)/yr it would yield 4.79/28.05*24.02=4.1018 Tg(C)/yr, which rounds up to 4.10, as quoted in the table. The same we assure for $C_2H_6$: $\gamma$=24.02/30.07=0.7988, 2.73*24.02/30.07=2.1807 Tg(C)/yr, or 2.18 rounded to 0.01. We use spreadsheet software accounting for the atomic content of the molecules to derive the conversion factors and assure that these are correct.

Table 5: Several points are not clear to me:

a) how is the aggregate uncertainty factor derived specifically?

We add a respective note.

b) What is the relation between columns 2,3,4 (individual surface sources) and column 6.

Column 6 is the sum of the columns 2,3,4 reduced from [Tg(gas)/yr] to [Tg(C)/yr] units, as we emphasise in the notes to the table. This helps to better compare the total trace C influx to the atmospheric reservoir.

c) Column 6 seems way too low as total surface emission for CO.

See our answer to b) above.

d) Can you comment on the huge error bar for the isoprene emissions?

This error bar (which of course makes sense as a forward estimate, capped by zero on the lower end) reflects large emission uncertainties (factors 3 and higher) associated with the biogenic sources, as we emphasise in the manuscript (L582).

Table 6: Is the uncertainty for CH3Cl isotopic composition really that low?

As stated, this is error of the mean from Thompson *et al.* (2002).

4) Part of the discussion (L630 ff) is very hard to follow and in my view does not reflect the results shown in the figures, see detailed points below. Another part is L672 ff (last paragraph of section 4.1). The conclusions that the authors are hard to related to the results shown in the figures.

We have amended the discussion part, with a great deal of help from the comments on scientific/presentation issues. Please, see our answers to these below.

5) The issue that inverse models report too optimistic posteriori errors for the combined source is quite a strong statement. I wonder whether this is not a misunderstanding by the authors. According to my knowledge inverse modeling involves rather solid error calculation, and especially in inverse modeling the constraint on the total source (from mole fraction observations) is much more tight than the sum of the individual components. I suggest that the authors contact with an inverse modeler to check this.

Thank you, we did; we are certain about this statement. First, it is based on the fundamental mathematical apparatus which is applicable to (*i.e.* analytically derivable from) the estimates conveyed by the regarded studies. Noteworthy, neither of the latter provides the uncertainty of the overall CO emission flux and $\delta^{13}C$, and we are not aware of reasons for that. Second, using any inverse modelling framework (here Bayesian estimation) *requires* the analysis of the posterior solution distribution, *e.g.*, via an analytical solution, a systematic study of cases or a Monte Carlo study (see the review on that in Enting, 2002, Sect. 3.2). As pointed out by Tarantola (2005) (Sect. 3.3), at least a trivial estimate of the uncertainties correlation is always possible. Third, since no such estimate is provided, we would like to infer the upper limit (the "worst case") of the final uncertainty, which is to be that of the correlated case. We add this elucidation to Sect. 2.2.

**Scientific and presentation issues:**

Title: This is in my view an uncommon usage of the term proxies. Does it need to be in the title?

This is a valid point. We suggest the title:
"Uncertainties of fluxes and $^{13}C/^{12}C$ ratios of atmospheric reactive gases emissions".

Line 27 "which factors determine a particular emission source isotope ratios". The authors are encouraged to avoid such dense constructions with multiple nouns. For a reader, it is much easier to grasp constructions with "of". In this case "which factors determine the isotope ratio of a particular

emission source". This could be simplified at many places throughout the paper: Here, there is also an "s" too much. Also, in the next sentence, please specify "the latter"

Ok, done

L47: see comment L27, too many nouns. . .

Ok, done

L58: "Tendencies" is a rather unspecific term. Why not concentrations?

L58/59: "by modifying its vertical diffusive flux boundary conditions at the lowest model

layer" is very technical. Change to "by adding emissions to the . . ."

We agree that this sentence is too technical and redundant. We remove it.

L86: For me "isotope separation" is a bit strange term in this context. Why not: Fluxes of individual isotopologues?

Ok, done

L 92: You could help the reader with an example calculation of Eq 1. E.g. for the case of CO with one 13C and one 18O atom. How do the ratios, deltas and fluxes behave?

As we note in our reply to Referee 1 above, index j was not explained in the manuscript, which causes confusion. In the revised manuscript, an example for $^{13}CO$ or $C^{18}O$ will be straightforward (we do not consider rare substitutions like $^{13}C^{18}O$).

L103-109: This is very hard to follow, please reformulate. Avoid "ex post facto".

We have changed this paragraph (also at the advice of Referee 1) but would like to keep "ex post facto"

L142 ff: please rewrite sentence that includes Eq 5.

Ok, done

L150 ff: If the rare isotopologue fluxes are off by 1% and the abundant isotopologue flux is correct, the isotope ratios would be wrong by 10 permil. Please comment/clarify.

We admit that discussion using Eqs. (5)–(6) is a too clever by half attempt to reach the textbook Eqs. (7)–(8). As we reply above, we have removed this part of Sect. 2.2. Furthermore, some formulae were erroneously typeset, we apologise for that.

L166: I cannot understand the remark on the uncertainties of guessed parameters. When you guess a parameter, you can often also guess an uncertainty.

Here we imply information that is not based on a measurement or derived via logical but not quantifiable conclusion, that is, an assumption. We reformulate as "Often uncertainties of assumptions (...) cannot be quantified using strict mathematical apparatus, hence should be analysed in a sensitivity framework."

L275: what about the sensitivity in other regions of the world (Africa, South America, Asia)?

Fair, we add the global and zonal averages.

L306: where does the ratio 250/280 come from?

It is bio- versus fossil fuels. We remove the parentheses in L306 for clarity.

L353: Cryptic sentence about the use of a different proxy for CO in GFED. Not clear to me

OK, reformulated.

L382 and Fig 6: It is not clear for me how you come from the -20.5 for the marine carbon content in the text to the -13 in Fig 6 for oceanic CO.

The choice of the different signature for CO is explained in the following sentences (*cf.* L385).

L 480/1 and Fig 6. It is not visible in Fig 6 that NH terrestrial sources are smaller than oceanic ones in winter. Fig 6 implies that oceanic sources are zero?

Fig. 6 shows that oceanic source is present within 1 Gg/yr (per grid cell) where the $\delta^{13}C$ value of $-13‰$ is defined. Changes in zonal average $\delta^{13}C$ between 30°N and 60° throughout October-March are seen (red-brownish shaded areas, reaches $-(15-13)‰$). We will use "comparable" instead of "weaker than".

L490: What causes the range in the isoprene emission?

We reformulate the sentence.

L501: No emissions in Africa in Fig 7 (see above).

See the reply to the comment 2) above.

L541: why is photolysis of chloromethane included as isotope resolved processes

Photolytic breakdown of $CH_3Cl$ produces methyl radical which quickly recombines with air $O_2$ to yield a methylperoxy radical. It is thus possible to account for this (minor) source of C entering the $CH_4 \rightarrow CO$ oxidation chain (assuming there are no significant KIEs in the $CH_3Cl$ photolysis). We change "decomposing" to "yielding".

L570/1: Either number or plant type is wrong in the example.

Thank you, corrected.

L582: Where do the UF estimates come from?

From Guenther *et al.* (1995), as referenced. We add this reference to the notes in Table 6 for clarity.

L592: Where does the biofuel uncertainty come from? A table, or is this additional information?

It is considered to be a $C_3/C_4$ plant composite. We add a clarification to the beginning of the section.

L605: do not let the reader guess which of the studies are bottom-up.

We add a footnote to the Table 7 which specifies which studies are "bottom-up", respectively.

Line 630 - 658: This part is not clear. I have a hard time following the arguments and finding back in the figures what the authors describe.

a) B00 does not really seem to have a much lower Ch4 derived CO source than B99 or SM89 (line 630)

Correct, here the *a posteriori* estimate is implied; we will amend the sentence. Nonetheless, we do not know the absolute CH₄-derived CO term from SW89.

b) the a posteriori sources . . . (line 633). The logic is wrong or at least not clear. I do not see that a posteriori sources are reduced in M97.

c) (l634) I do not see that B00 decreases CH4 derived CO less than M97. In the posteriori results the BB source also increases. The description is not clear at all, and it is also not clear what the bottom line should be.

We have amended the sentence (also, B00 was erroneously exchanged with M97).

d) L635: These two studies do NTO show the largest BB emissions, this is the case for SW89.

The estimate of SW89 is rather uncertain w.r.t. the source apportioning, so we prefer not to use it for individual source magnitude comparison. We add an elucidation above and amend the sentence (also, B00 was erroneously exchanged with M97).

e) L645: please help the reader why and how the cold start issue could be addressed by 18O but not by 13C

OK, done

f) L646: It is not immediately clear that the strengths of other sources can be constrained better with isotopes.

We do not state that.

**Technical issues:**

L29-30: . . . how comprehensive should the model be

OK, done

L46: It is not clear what "in the evaluation setup" means. Is this necessary?

No comment

L61: at THE respective

OK, done

L67: specific emission category

OK, done

L74: delete "THE"

OK, done

L75: simplify: . . .. , which leads to more realistic. . ..

OK, done, but kept two sentences

L79: the OBSERVED mixing

OK, done

L87 regular ! total

OK, done

L114: not sure what these preparation tools are. Leave out?

Want to keep

L115: avoid double plurals (fluxes values –? flux values), also in several other places.

OK, done, most of the times

L124: leave out "superposed"

OK, done

L125: clearly comprehensible ! clear

OK, done

L126: or various isotope mixtures ! with different isotopic composition

OK, done

L127: abundance ! source strength

We talk about summing compartments here, which can be turned into fluxes by relating them to the unit it time.

L129: Leave out "To give an example". This is not an example.

OK, done

L136: avoid "impermeable"

We like to keep this not common but useful expression. Not until AI writes paper to AI papers will be free of human induced peculiarities in formulations.

L136/7: "because in contrast to ratios, it is much more difficult to relate" ! because it

is difficult to relate

OK, done

L164: rewrite/explain "by fitting their (isotope mass-balanced) sum to the given integral."

OK, we use "distribute the shares" instead of "fit the sum".

L168 with ! using

OK, done

L176: add ISOTOPE ratio

OK, done

L186/7. Leave out this sentence, it creates more confusion that clarification. (Why would it?)

Thank you, we agree here, this is straightforward from definition of $^iR_e$.

L188 the ! a

OK, done

L202: "surface and adjacent layers" is unspecific. I think you mean the first and second model layer, correct?

The number of the layer depends on the vertical resolution of the model, therefore we use "adjacent layers". This pertains to the following sentence as well.

L203: specify the remaining sectors

These are power generation, industrial fuel usage and waste treatment sectors mentioned above. We reformulate these two sentences.

L206: and OTHER emitted

OK, done

L262: significantLY HIGHER

OK, done

L272: define "bio-petrol"

Apologies, this s a somewhat unfinished edit. Of course, implied is *less* extensive use of biofuel in EU and NA.

L315: use different word for "superincumbent" ! higher?

Ok, we now use "overlying layers"

L320 verb missing (is)

OK

L329: second MODEL layer

Correct, we use "near-surface" model layer.

L338: El Nino Southern Oscillation (ENSO) climate pattern

Changed

L341/2: The variation of the d13C of the emission flux is. . .

OK, done

L401: replace "tolerating"

We think "tolerating" is correct and clear

L436: replace "escorted"

OK, done

L437: replace "rational"

We think "rational" is correct and clear

L445/6: Rewrite: "may be used as a proxy for the average bulk leaf biomass, thus concluding the depletion of the emitted isoprene in relation to it."

OK, done

L459: framework DEVELOPED by

OK, done

L461: a set of numerous parameters

OK, done

L477: replace perceptibly by more quantitative term, or leave out

OK, changed

L478: . . . which results in

OK, done

L495: Avoid double plural (trace gas emissions)

OK, done

L501 & 503: the largest . . .. A comparable. Please modify.

OK, done

L530: . . .sources associated with biogenic activity that emit isotopically light methane

OK, done

L531: corresponding TO

OK, done

L540: replace "isotopic carbon"

OK, done

L562: In contrast, uncertainties of isotope signatures are reported . . .

OK, done

L565/6; rewrite sentence

OK, done

L583: avoid double plural

OK, done

L620: One infers a similar. . .

OK, done

L650: Replace/leave out "inquiries"

OK, done

**References**

[revised manuscript text omitted]
^{\text{th}}$ rare isotopologue influx $^{\text{rare},i}F$ to the (regular) emission flux $F$ as

$$^iR = \frac{^{\text{rare},i}F}{q \cdot {}^{\text{abun}}F + (q-1)\sum_j {}^{\text{rare},j}F} \simeq \frac{^{\text{rare},i}F}{q \cdot F \cdot \left(1 + 2(q-1)\sum_j {}^jR_{st}\right)}, (5)$$

assuming that the fraction of the rare isotopologues is negligibly small in the total flux, which is valid for the isotopes of the light elements (*e.g.* C, N, O). This is the only approximation that affects the further analysis. Neglecting the abundant isotopes in the rare isotopologues introduces errors in the estimate of $F$ on the order of 1 % for carbonaceous species, assuming an average fraction of $^{13}$C carbon of 1 % in the total flux. Thus the resulting approximation of the flux

$$^{\text{rare},i}F \simeq {}^iR \cdot q \cdot F \cdot \left(1 + 2{}^iR_{st}(q-1)\right) \quad (6)$$

is approximately 1 % inaccurate for CO and 5 % for isoprene ($C_5H_8$), *i.e.* depending on the number of carbon atoms incorporated in the species molecule. Compared to the typically large errors for the emission fluxes (see below), this inaccuracy is an order of magnitude smaller.

**Page 5: [2] Deleted**       **Sergey Gromov**       **29/04/2017 1:34 PM**

. Clearly then, the resulting total flux isotopic ratio $^iR_e$ is

$$^iR_e = \varphi \sum_s {}^iR_s \cdot F_s, \qquad \varphi \equiv \left(\sum_s F_s\right)^{-1} \quad (8)$$

Here